# Provable Memorization Capacity of Transformers

**Junghwan Kim**
CSE Department
University of Michigan
Ann Arbor, MI
kimjhj@umich.edu

**Michelle YoungJin Kim**
CSE Department
Michigan State University
East Lansing, MI
kimmic16@msu.edu

**Barzan Mozafari**
CSE Department
University of Michigan
Ann Arbor, MI
mozafari@umich.edu

## Abstract

Quantifying memorization capacity is essential for understanding the expressiveness and generalizability of deep learning model architectures. However, the memorization capacity of the Transformer architecture has yet to be explored. In this work, we present the first study of the memorization capacity of the Transformer architecture. We prove that Transformers are capable of memorizing $N$ sequence-to-sequence mappings of length $n$ with $d$-dimensional input tokens using $\tilde{O}(d + n + \sqrt{nN})$ parameters. Our theory supports memorization both with and without permutation equivariance, utilizing positional encodings in the latter case. Building on our theory, we also analyze the memorization capacity of Transformers in the sequence classification and language modeling tasks. To verify these theoretical findings, we conduct experiments analyzing the memorization capacity of Transformers in the natural language domain.

## 1 Introduction

Transformer networks (Vaswani et al., 2017) have shown tremendous success in natural language processing tasks (Devlin et al., 2019; Yang et al., 2019; Brown et al., 2020; Fedus et al., 2022), rapidly becoming the standard architecture for natural language modeling. The success of Transformers has also transferred to various other sequence and set modeling tasks, including image recognition (Parmar et al., 2018; Dosovitskiy et al., 2021), semantic segmentation (Zheng et al., 2021), video understanding (Akbari et al., 2021; Bertasius et al., 2021), reinforcement learning (Parisotto et al., 2020; Chen et al., 2021; Janner et al., 2021), 3D point cloud processing (Zhao et al., 2021), protein structure prediction (Jumper et al., 2021), and automatic theorem proving (Polu & Sutskever, 2020). Despite this success across various areas, the theoretical understanding of Transformers lags behind that of standard fully-connected networks.

The major strength of Transformers is in their efficient scaling, which is enabled through parallel token processing with parameter sharing and simple dot-product-based token interaction. Surprisingly, even though the parameter sharing and simple token interaction impose constraints on the function space of Transformers, Yun et al. (2020a) show that Transformers can approximate any continuous function from input to output sequences. However, their result focuses on the function approximation capacity with infinite precision, leaving the finite sample memorization capacity with finite precision unexplored. We note that universal function approximation does not automatically imply efficient memorization in terms of the number of parameters. Generalizing infinite precision results to the finite precision case is not straightforward and may not be possible in some cases. For example, Transformers are Turing complete only with infinite precision (Pérez et al., 2019), but not with finite precision (Dehghani et al., 2019).

Understanding the memorization capacity of a model is critical for choosing an appropriate model size. Practitioners often choose a model size with enough representation capacity to achieve zero training loss (i.e., a size larger than memorization capacity). Moreover, the memorization capacity has generalization implications, as observed in the double descent phenomena (Belkin et al., 2019; Nakkiran et al., 2021). As the network size increases, generalization performance exhibits a bias-variance tradeoff until the memorization is possible and then improves monotonically after-

ward. Understanding the memorization capacity of Transformers requires answers to the following questions: How large should the size and precision of the Transformer architecture be to enable memorization of any given number of input-output sequence pairs? How does the memorization capacity of Transformers differ for various problem settings in practical application scenarios?

In this paper, we answer these questions by proving that Transformers can memorize $N$ sequences of $d$-dimensional tokens with length $n$ using $\tilde{O}(d + n + \sqrt{nN})$ parameters. Our proof constructs permutation equivariant Transformers that can memorize all permutations of $N$ input sequences. We extend this construction to the memorization without permutation equivariance by adding positional encodings. In addition, we derive the memorization capacity for sequence classification task from our proposed theory.

The key technical component of our construction is efficient contextual mapping, which requires only $n$ self-attention layers. Our contextual mapping also applies to sparse-attention Transformers making fewer assumptions on sparsity patterns than Yun et al. (2020b) and Zaheer et al. (2020). Furthermore, we present the generalization of contextual mapping to function approximation settings, vastly improving the parameter efficiency of attention layers compared to the selective-shifting-based contextual mapping in Yun et al. (2020a).

Our main contributions are summarized as follows:

- We prove the memorization capacity of Transformers for sequence-to-sequence mappings with and without permutation equivariance. We analyze the memorization capacity in other standard task settings, such as sequence classification and language modeling.

- We show that the efficient contextual mapping presented in our theoretical analysis extends to sparse attention settings and improves the function approximation results.

- We provide experiments validating the memorization capacity of Transformers for token classification and sequence classification tasks.

## 1.1 Related works

**Memorization capacity.**    Characterizing the memorization capacity of neural networks has been an active research area with a long history (Baum, 1988; Sontag, 1997; Huang & Babri, 1998; Huang, 2003; Zhang et al., 2017; Yun et al., 2019; Bubeck et al., 2020; Vershynin, 2020; Rajput et al., 2021; Park et al., 2021; Vardi et al., 2022). Recently, Park et al. (2021) constructed neural networks with $O(N^{2/3})$ parameters to memorize $N$ data points. They bypass the $\Omega(N)$ lower bound in Sontag (1997) by assuming a simple separation (i.e., $\|\mathbf{x}_i - \mathbf{x}_j\| \geq \delta, \forall i \neq j$). Vardi et al. (2022) improve further, showing that $\tilde{O}(N^{1/2})$ parameters are sufficient. They also prove the matching lower bound of $\tilde{\Omega}(N^{1/2})$ through the VC-dimension analysis.

Inspired by Park et al. (2021) and Vardi et al. (2022), our construction assumes a similar separation, but it is between pairs of distinct tokens, not the whole sequence pairs. (See Definition 3.1 and the discussion that follows the definition.) In addition, our construction uses the same pipeline of projection, string matching, and bit extraction as in Vardi et al. (2022). However, we introduce an additional critical step: efficient contextual mapping to complement projection by summarizing all token information via self-attention layers.

In contrast to the extensive results on fully-connected networks, there are few studies on the memorization capacity of specific modern architectures. Hardt & Ma (2017) show that the residual network with ReLU activation and $O(N)$ hidden neurons can memorize $N$ data points under the separation assumption. Nguyen & Hein (2018) show that the convolutional network with $O(N)$ hidden neurons can memorize $N$ data points. To the best of our knowledge, there is no existing literature on the memorization capacity of the Transformer architecture.

**Transformer expressivity.**    Given the recent empirical success of Transformers observed across multiple areas, several papers have studied the expressivity of Transformers. Yun et al. (2020a) establish the first universal approximation theorem for Transformers, and the result is later extended to sparse-attention Transformers (Yun et al., 2020b; Zaheer et al., 2020) and Transformers with hard constraints (Kratsios et al., 2022). All these results study the function approximation but not

the finite sample memorization as in our paper. We also note that our construction can reduce the number of self-attention layers in the function approximation setting.

There are other lines of studies focusing on different aspects of the representation capacity of Transformers. Some papers aim to characterize the representation capacity of a single self-attention layer. Bhojanapalli et al. (2020) suggest that the small size of attention heads limits the rank of a self-attention matrix. Dong et al. (2021) show that the rank of self-attention decays exponentially when self-attention layers are composed without skip-connections or feedforward layers. Likhosherstov et al. (2021) show that a fixed self-attention module can approximate any sparsity pattern.

Other papers investigate a tradeoff between width and depth since it is a crucial issue when scaling Transformers. Levine et al. (2020) demonstrate the depth efficiency in modeling feature interaction through the separation rank analysis of Transformers. Wies et al. (2021) identify the rank of the input embedding matrix as a bottleneck for the network width's contribution to expressivity.

Finally, Vuckovic et al. (2020) and Kim et al. (2021) study the Lipschitz smoothness of attention operations. Edelman et al. (2022) derive the norm-based generalization bounds from the Lipschitz smoothness of norm-bounded attention layers to analyze the inductive bias of attention layers. Wei et al. (2021) study the expressivity of Transformers under the constraint of statistical learnability. Our memorization study provides a complementary understanding of the capacity of Transformers.

## 2 PRELIMINARIES

This section establishes the notation and defines the Transformer architecture.

### 2.1 NOTATION

Denote the number of input-output pairs as $N$, the number of output classes as $C$, the token embedding dimension as $d$, and the sequence length as $n$. We use $\tilde{O}(\cdot)$ to hide logarithmic factors and $O(\cdot)$ to hide constant factors. We use $\sigma_R$ to represent the ReLU activation function. We let $\sigma_S$ and $\sigma_H$ be the softmax and hardmax operators, respectively. These operators take a matrix as an input, apply softmax/hardmax columnwise, and output a column stochastic matrix of the same size.

For $m \in \mathbb{N}$, we define $[m] = \{1, 2, \cdots, m\}$. We use $|\mathcal{S}|$ to denote the number of elements in a set $\mathcal{S}$. We denote a set or a function as an upper case caligraphic letter, a matrix by an upper case bold letter and a vector by a lower case bold letter. We denote the standard unit vector with all but $i$-th coordinates 0 as $\boldsymbol{e}_i$, the $m$ dimensional vector with all coordinates 1 as $\mathbf{1}_m$ and the $m$ dimensional vector with all coordinates 0 as $\mathbf{0}_m$. For a vector $\boldsymbol{x}$, we represent its $i$-th entry as $\boldsymbol{x}[i]$ and its Euclidean norm as $\|\boldsymbol{x}\|$. For a matrix $\boldsymbol{X}$, we use $\boldsymbol{X}[i, j]$, $\boldsymbol{X}[i, :]$, and $\boldsymbol{X}[:, j]$ to represent $(i, j)$-th entry, $i$-th row, and $j$-th column, respectively. We use $\|\boldsymbol{X}\|_F$ to represent the Frobenius norm of the matrix $\boldsymbol{X}$.

### 2.2 TRANSFORMER ARCHITECTURE

We define a Transformer $\mathcal{N} : \mathbb{R}^{d \times n} \to \mathbb{R}^{1 \times n}$ of depth $L$ as a composition of $L$ Transformer blocks with input and output embedding mappings:
$$\mathcal{N} = \mathcal{E}_{out} \circ \mathcal{F}_L \circ \cdots \circ \mathcal{F}_2 \circ \mathcal{F}_1 \circ \mathcal{E}_{in}$$
where each Transformer block $\mathcal{F}_l : \mathbb{R}^{m \times n} \to \mathbb{R}^{m \times n}$ is a sequence-to-sequence function consisting of two subblocks: a self-attention subblock and a tokenwise feedforward subblock. The input embedding block $\mathcal{E}_{in} : \mathbb{R}^{d \times n} \to \mathbb{R}^{m \times n}$ and the output embedding block $\mathcal{E}_{out} : \mathbb{R}^{m \times n} \to \mathbb{R}^{1 \times n}$ are 1-layer tokenwise linear mappings.

The self-attention subblock represents the interaction among tokens. Formally, given an input $\boldsymbol{Z} \in \mathbb{R}^{m \times n}$, the self-attention subblock $\mathcal{F}_l^{(SA)}$ with $h$ heads and head size $k$ computes

$$\mathcal{F}_l^{(SA)}(\boldsymbol{Z}) = \boldsymbol{Z} + \sum_{i=1}^h \boldsymbol{W}_{l,i}^{(O)} \left( \boldsymbol{W}_{l,i}^{(V)} \boldsymbol{Z} \right) \sigma_S \left[ \left( \boldsymbol{W}_{l,i}^{(K)} \boldsymbol{Z} \right)^T \left( \boldsymbol{W}_{l,i}^{(Q)} \boldsymbol{Z} \right) \right],$$

where $\boldsymbol{W}_{l,i}^{(O)} \in \mathbb{R}^{m \times k}$ and $\boldsymbol{W}_{l,i}^{(V)}, \boldsymbol{W}_{l,i}^{(K)}, \boldsymbol{W}_{l,i}^{(Q)} \in \mathbb{R}^{k \times m}$ are the weight matrices parametrizing the self-attention subblock. We include a skip-connection in the self-attention subblock.

The feedforward subblock processes each token independently in parallel by applying two feed-forward layers. Given an input $\boldsymbol{H} \in \mathbb{R}^{m \times n}$, the feedforward subblock $\mathcal{F}_l^{(FF)}$ with dimension $q$ computes

$$\mathcal{F}_l^{(FF)}(\boldsymbol{H}) = \boldsymbol{H} + \boldsymbol{W}_l^{(2)} \sigma_R \left( \boldsymbol{W}_l^{(1)} \boldsymbol{H} + \boldsymbol{b}_l^{(1)} \mathbb{1}_n^T \right) + \boldsymbol{b}_l^{(2)} \mathbb{1}_n^T,$$

where $\boldsymbol{W}_l^{(2)} \in \mathbb{R}^{m \times q}$, $\boldsymbol{b}_l^{(2)} \in \mathbb{R}^m$, $\boldsymbol{W}_l^{(1)} \in \mathbb{R}^{q \times m}$ and $\boldsymbol{b}_l^{(1)} \in \mathbb{R}^q$ parametrize the feedforward subblock. The feedforward subblock also includes a skip-connection.

Finally, the Transformer block composes two subblocks as

$$\mathcal{F}_l(\boldsymbol{Z}) = \mathcal{F}_l^{(FF)}(\mathcal{F}_l^{(SA)}(\boldsymbol{Z}))$$

Unlike the original formulation in Vaswani et al. (2017), our definition excludes layer normalization as Yun et al. (2020a) to simplify our analysis. Since layer normalizations mainly contribute to optimization without much effect on expressivity, our definition still captures the representation aspect of the Transformer architecture.

Since each Transformer block consists of a fixed number of layers even in the most fine-grained sense, we use the number of blocks $L$ as the *depth* of the network. We define the *width* of the network as $\max\{m, kh, q\}$[1]. The *number of parameters* is the number of non-zero weights in our network. We note that the single parameter is reused $n$ times for a sequence length $n$, but still is counted as one parameter. The *bit complexity of the network* is the maximum bit complexity of its weights, where the *bit complexity of a weight* is the number of bits required to represent it. We adopt these definitions of the number of parameters and the bit complexity from the convention in the VC dimension literature (Bartlett et al., 2019) and the recent paper on the optimal memorization capacity of fully-connected networks (Vardi et al., 2022).

## 3 MEMORIZATION CAPACITY OF TRANSFORMERS

In this section, we describe the problem setting and present our main theorem on the memorization capacity of the Transformer architecture. Then, we sketch the proof for our main theorem and discuss the memorization capacity of Transformers in other standard task settings.

### 3.1 PROBLEM SETTING

We consider the memorization of $N$ input-output sequence pairs $(\boldsymbol{X}^{(1)}, \boldsymbol{Y}^{(1)}), \cdots, (\boldsymbol{X}^{(N)}, \boldsymbol{Y}^{(N)})$ where each input $\boldsymbol{X}^{(i)} \in \mathbb{R}^{d \times n}$ is a sequence of $n$ token vectors in dimension $d$. Each output $\boldsymbol{Y}^{(i)} \in [C]^{1 \times n}$ is a sequence of $n$ labels where each label $\boldsymbol{Y}^{(i)}[1, k]$ is assigned to a token $\boldsymbol{X}^{(i)}[:, k]$. We define the *context* of each input sequence $\boldsymbol{X}^{(i)}$ as $\mathcal{V}^{(i)} = \{\boldsymbol{v} \in \mathbb{R}^d : \boldsymbol{v} = \boldsymbol{X}^{(i)}[:, k] \text{ for some } k \in [n]\}$. We define the *vocabulary* $\mathcal{V} = \bigcup_{i \in [N]} \mathcal{V}^{(i)}$ as the set of all tokens appearing in the input sequences. Note that $|\mathcal{V}| \leq nN$.

As pointed out in Park et al. (2021) and Vardi et al. (2022), we must assume some conditions on the dataset to bypass the lower bound in Sontag (1997) and memorize $N$ data points with $o(N)$ parameters. We present a natural generalization of the separation condition defined in Vardi et al. (2022) to sequence modeling settings.

**Definition 3.1.** Let $r \geq 1, 0 < \delta \leq 1$. Let $\boldsymbol{X}^{(1)}, \cdots, \boldsymbol{X}^{(N)} \in \mathbb{R}^{d \times n}$ be N input sequences with vocabulary $\mathcal{V}$. We say that $\boldsymbol{X}^{(1)}, \cdots, \boldsymbol{X}^{(N)}$ are tokenwise $(r, \delta)$-separated if

1. $\|\boldsymbol{v}\| \leq r$ for all $\boldsymbol{v} \in \mathcal{V}$ and
2. $\|\boldsymbol{v} - \boldsymbol{v}'\| \geq \delta$ for all $\boldsymbol{v}, \boldsymbol{v}' \in \mathcal{V}$ with $\boldsymbol{v} \neq \boldsymbol{v}'$.

This condition requires (1) each token to have a bounded norm and (2) each pair of distinct tokens to be separated. We note that the tokenwise separation is a stronger condition than the separation of input sequences[2]. However, the condition better captures many practical settings where the number of tokens in the vocabulary is much smaller than the number of input sequences.

---

[1] We recall that $m$ is the embedding dimension, $h$ is the number of attention heads, $k$ is the attention head size, and $q$ is the feedforward dimension.

[2] When $\boldsymbol{X}^{(1)}, \cdots, \boldsymbol{X}^{(N)} \in \mathbb{R}^{d \times n}$ are distinct and $(r, \delta)$-separated, then (1) $\|\boldsymbol{X}^{(i)}\|_F \leq r\sqrt{n}$ for $i \in [N]$ and (2) $\|\boldsymbol{X}^{(i)} - \boldsymbol{X}^{(j)}\|_F \geq \delta$ for $i, j \in [N]$ with $i \neq j$.

For the permutation equivariant mappings, we need the following label consistency condition.

**Definition 3.2.** Let $(\boldsymbol{X}^{(1)}, \boldsymbol{Y}^{(1)}), \cdots, (\boldsymbol{X}^{(N)}, \boldsymbol{Y}^{(N)}) \in \mathbb{R}^{d \times n} \times [C]^{1 \times n}$ be $N$ input-output pairs of sequences. We say that $(\boldsymbol{X}^{(1)}, \boldsymbol{Y}^{(1)}), \cdots, (\boldsymbol{X}^{(N)}, \boldsymbol{Y}^{(N)})$ are consistently labeled if

$$\boldsymbol{X}^{(i)}[:, k] = \boldsymbol{X}^{(i)}[:, l] \text{ implies } \boldsymbol{Y}^{(i)}[1, k] = \boldsymbol{Y}^{(i)}[1, l]$$

for every $i \in [N]$ and $k, l \in [n]$.

We emphasize that we impose this condition only on the memorization with permutation equivariance but not on the memorization without permutation equivariance[3]. The condition implies that two identical tokens appearing in the same context should have the same label. Consider a permutation equivariant mapping $\mathcal{F} : \mathbb{R}^{d \times n} \to \mathbb{R}^{1 \times n}$ and an input sequence $\boldsymbol{X} \in \mathbb{R}^{d \times n}$. Define $\boldsymbol{X}'$ by swapping two tokens $\boldsymbol{X}[:, k]$ and $\boldsymbol{X}[:, l]$ in $\boldsymbol{X}$. Then, we have $\mathcal{F}(\boldsymbol{X})[1, k] = \mathcal{F}(\boldsymbol{X}')[1, l]$ due to the permutation equivariance. However, if two tokens $\boldsymbol{X}[:, k]$ and $\boldsymbol{X}[:, l]$ were identical, then $\boldsymbol{X} = \boldsymbol{X}'$ and consequently $\mathcal{F}(\boldsymbol{X})[1, k] = \mathcal{F}(\boldsymbol{X})[1, l]$.

### 3.2 MAIN RESULTS

We now present our main theorem on the memorization capacity of Transformers.

**Theorem 3.1.** *Let $N, d, n, C \in \mathbb{N}$ and $r \geq 1, 0 < \delta \leq 1$. Let $(\boldsymbol{X}^{(1)}, \boldsymbol{Y}^{(1)}), \cdots, (\boldsymbol{X}^{(N)}, \boldsymbol{Y}^{(N)}) \in \mathbb{R}^{d \times n} \times [C]^{1 \times n}$ be $N$ input-output sequence pairs of sequences where input sequences are distinct and tokenwise $(r, \delta)$-separated.*

1. *(With permutation equivariance) Suppose that contexts $\mathcal{V}^{(i)}$ are distinct and sequences are consistently labeled. Then, there exists a Transformer network $\mathcal{N} : \mathbb{R}^{d \times n} \to \mathbb{R}^{1 \times n}$ such that*

$$\mathcal{N}(\boldsymbol{X}^{(i)} \boldsymbol{P}) = \boldsymbol{Y}^{(i)} \boldsymbol{P}$$

   *for every $i \in [N]$ and for every permutation matrix $\boldsymbol{P} \in \mathbb{R}^{n \times n}$.*

2. *(Without permutation equivariance) There exists a Transformer network $\mathcal{N} : \mathbb{R}^{d \times n} \to \mathbb{R}^{1 \times n}$ and positional encoding $\boldsymbol{E} \in \mathbb{R}^{d \times n}$ such that*

$$\mathcal{N}(\boldsymbol{X}^{(i)} + \boldsymbol{E}) = \boldsymbol{Y}^{(i)}$$

   *for every $i \in [N]$.*

*In both cases, the Transformer $\mathcal{N}$ has width 16 ($m = 8$, $h = k = 1$ and $q = 16$), depth*

$$O\left(n + \sqrt{nN \log(nN)} + \sqrt{\frac{nN}{\log(nN)} \cdot \max\{\log C, \log R\}}\right)$$

*and bit complexity bounded by*

$$O\left(\log d + \sqrt{\frac{nN}{\log(nN)} \cdot \max\{\log C, \log R\}}\right)$$

*where we denote $R := 8000r^2 \delta^{-2} dn^5 N^6$.*

Theorem 3.1 shows that $\tilde{O}(d + n + \sqrt{nN})$ parameters are enough to memorize $N$ sequence-to-sequence mapping of length $n$ with token dimension $d$ since the initial embedding layer has $d$ parameters and the rest layers have a constant number of parameters. We provide the proof sketch in Section 3.3 and the full proof in Appendix A.

**Remark 3.2.** *Extensions to real vector outputs.* Some application scenarios of Transformers require real vector outputs. As proposed in Park et al. (2021) and Vardi et al. (2022), extension to real scalar values is easily achievable by using $O(\frac{1}{\epsilon})$ classes when the output has a bounded range. More concretely, we partition the output range into $\epsilon$-length intervals and match each class to one partition to perform regression with error $\epsilon$ per token. This replaces $\log C$ in Theorem 3.1 with $\log(\frac{1}{\epsilon})$.

---

[3]The general sequence-to-sequence mappings do not satisfy the condition. For example, the same word appearing multiple times in a sentence may have different meanings.

Similarly, extension to vector outputs is possible when the output has a bounded domain. Suppose that we aim to minimize the tokenwise L2 distances in dimension $p$. We partition the output range into $\frac{\epsilon}{\sqrt{p}}$-length cubes and match each class to one cube. Then, we use $O\left(\left(\frac{\sqrt{p}}{\epsilon}\right)^p\right)$ classes to perform regression with error $\epsilon$ per token. This construction replaces $\log C$ in Theorem 3.1 with $p \log(\frac{p}{\epsilon})$.

**Remark 3.3.** *Large width and fixed bit complexity.* Theorem 3.1 uses fixed width and large bit complexity to minimize the number of parameters. However, a common approach to scaling Transformers is to increase width (Levine et al., 2020) while using the same number of bits per parameter. Using a similar argument as Vardi et al. (2022), we extend Theorem 3.1 to the cases with a larger width and with bounded bit complexity. When the larger width is allowed, Transformers of width $O(nN/L^2)$, depth $\tilde{O}(n + L)$ and bit complexity $\tilde{O}(L)$ memorize the same dataset for some $L \leq \sqrt{nN}$. When the bit complexity is bounded, Transformers of width $O(1)$, depth $\tilde{O}(n+nN/B)$ and bit complexity $\tilde{O}(B)$ memorize the same dataset for some $B \leq \sqrt{nN}$. We provide the formal theorem and the proof in Appendix B.

**Remark 3.4.** *Tightness in the order of bit counts.* Suppose the token dimension $d$ and the sequence length $n$ are both $O(N)$. Theorem 3.1 shows that a Transformer memorizes $N$ input-output sequence pairs using $\tilde{O}(\sqrt{nN})$ parameters of bit complexity $\tilde{O}(\sqrt{nN})$, which sum up to $\tilde{O}(nN)$ bits. Without any additional assumption on a dataset, representing models that memorize all $C^{nN}$ possible labels[4] for $N$ input sequences requires $\Omega(nN)$ bits. Thus, Theorem 3.1 is tight upto logarithmic factors in the order of bit counts.

**Remark 3.5.** *Comparison against fully-connected ReLU networks.* With a slight modification of results in Vardi et al. (2022)[5], fully-connected ReLU networks require $\tilde{O}(dn + \sqrt{nN})$ parameters.

- The dependence on the number of data points $N$ is the same as $\tilde{O}(\sqrt{nN})$. However, for the permutation equivariant case, Transformers memorize all permutations of each input sequence. That is, Transformers are capable of memorizing at most $n!$ times more data points at the cost of reusing each parameter $n$ times[6].

- Our construction has a better dependence on $d$ and $n$: $O(d+n)$ for Transformers and $O(dn)$ for fully-connected ReLU networks. Transformers exploit the structure of sequence data through parameter sharing. As a result, Transformers do not need $O(dn)$ parameters to read all $dn$ values in the input sequence.

We note that this comparison is not completely fair because (1) our result makes a slightly stronger assumption of the separation between tokens than the separation between whole input; and (2) fully-connected networks are not designed for permutation equivariant datasets. However, the difference in assumption affects the final bound only to the logarithmic terms. Furthermore, there is no known result on the memorization capacity of other permutation equivariant architectures.

## 3.3 PROOF SKETCH

We outline the proof of Theorem 3.1. A more formal statement of each stage with detailed proof is in Appendix A. Our proof adopts the approach from Vardi et al. (2022) and shares similar steps. We discuss our technical novelty after sketching the main ideas of our proof.

Our proof constructs a Transformer in 4 stages. The first two stages assemble input values and encode as a "contextual token id" that identify each token in the different context of sequence. We ensure that the "contextual token id" is permutation equivariant and that the "contextual token id" is uniquely assigned to each token in each context. Then, the last two stages map each "contextual token id" to the corresponding label using the bit-extraction network adopted from Vardi et al. (2022). We describe key ideas of each stage.

---

[4]There are $C$ possible labels for $n$ tokens in $N$ sequences.

[5]We consider inputs as $dn$-dimensional flattened vectors and outputs as values from $[C^n]$. We choose a different balancing point between stages 2 and 3 in their construction to balance the extra factors.

[6]A similar benefit of Transformers has been previously observed in the function approximation (Yun et al., 2020a): the number of parameters is reduced by $(n-1)!$ times to successfully approximate permutation equivariant functions.

- **Stage 1. Tokenwise Projection.** We project each token vector to a scalar token id while keeping distinct tokens well separated.
- **Stage 2. Contextual Mapping.** We compose a sequence id as a linear combination of token ids. Weights of each id in the linear combination depend on the order of token id within each sequence. The resulting sequence ids are permutation invariant. [7] We concatenate the token id and the sequence id to obtain a contextual token id.
- **Stage 3. String Lookup.** We partition all $nN$ contextual token ids into intervals, each containing the same number of ids. We construct two encoding numbers for each interval by concatenating all corresponding contextual token ids and token labels. Then, we find which group the contextual token id falls into and retrieve the corresponding encoding numbers.
- **Stage 4. Bit Extraction.** We extract each contextual token id and token label from the encoding numbers. If the extracted contextual token id agrees with the one composed in stage 2, then we output the corresponding token label.

Our technical novelty in this proof is in (1) the implementation of the contextual mapping in stage 2 using self-attention subblocks and (2) generalizing stages 3 and 4 to incorporate skip connections.

**Remark 3.6.** *Contextual mapping.* The number of self-attention subblocks that our proof uses is proportional to the length of the sequence $n$ but independent of the number of data points $N$. This requirement is in striking contrast to the selective-shifting-based contextual mapping from Yun et al. (2020a), which requires $\left(\frac{1}{\delta}\right)^{dn}$ layers for shifting each grid cell of side length $\delta$. In the memorization setting, we may remove layers for grid cells without any data point. But the selective-shifting-based contextual mapping would still need $nN$ self-attention subblocks, which alone is already larger than the number of required layers in our result. In contrast, our efficient contextual mapping uses $n$ layers, improving all the above when $\delta, N > 1$. [8] We showcase the benefit of our contextual mapping in Appendix C. Specifically, we show that our contextual mapping are capable of incorporating sparse self-attention settings with minimal parameter overhead. Moreover, we show that the same idea of our contextual mapping can be applied in the function approximation setting.

**Remark 3.7.** *Parameter contribution.* In our construction, self-attention subblocks contribute only $O(n)$ parameters while feedforward subblocks contribute $O(\sqrt{nN})$. When $n < N$, most of the parameter count comes from feedforward subblocks. Although the self-attention layers play a critical role in contextual mapping, we do not need many of them. This explains the model design in practice, where more than half of the parameters are in the tokenwise feedforward subblocks (Vaswani et al., 2017; Devlin et al., 2019; Brown et al., 2020; Geva et al., 2021). Moreover, Mandava et al. (2020) observes that the self-attention layers can be further reduced without much performance loss.

## 4 MEMORIZATION CAPACITY IN OTHER TASKS

In this section, we generalize Theorem 3.1 to analyze the memorization capacity for other standard task settings. We consider sequence classification in Theorem 4.1, masked language modeling in Theorem D.1 and autoregressive language modeling in Theorem D.2. Due to the limited space, formal results on language modeling tasks and the proofs of the theorems are provided in Appendix D.

### 4.1 SEQUENCE CLASSIFICATION

In sequence classification, we assign a single label $y^{(i)} \in [C]$ for each input sequence $\boldsymbol{X}^{(i)}$. We present our theorem for sequence classification task.

**Theorem 4.1.** *Let $N, d, n, C \in \mathbb{N}$ and $r \geq 1, 0 < \delta \leq 1$. Let $(\boldsymbol{X}^{(1)}, y^{(1)}), \cdots, (\boldsymbol{X}^{(N)}, y^{(N)}) \in \mathbb{R}^{d \times n} \times [C]$ be $N$ input-output pairs of sequences where input sequences are distinct and tokenwise $(r, \delta)$-separated.*

1. *(With permutation invariance) Suppose that contexts $\mathcal{V}^{(i)}$ are distinct. Then, there exists a Transformer network $\mathcal{N} : \mathbb{R}^{d \times n} \to \mathbb{R}^{1 \times n}$ such that*

$$\mathcal{N}(\boldsymbol{X}^{(i)}\boldsymbol{P})[1, k] = y^{(i)}$$

---

[7] An appropriate choice of positional encoding bypasses permutation invariance by collecting token ids in the position order. See the second part of Theorem 3.1 for the result and Section A.5 for the details.

[8] The most practical settings fall into this regime.

*for every $i \in [N], k \in [n]$ and for every permutation matrix $\boldsymbol{P} \in \mathbb{R}^{n \times n}$.*

2. *(Without permutation invariance) There exists a Transformer network $\mathcal{N} : \mathbb{R}^{d \times n} \to \mathbb{R}^{1 \times n}$ and positional encoding $\boldsymbol{E} \in \mathbb{R}^{d \times n}$ such that*

$$\mathcal{N}(\boldsymbol{X}^{(i)} + \boldsymbol{E})[1, k] = y^{(i)}$$

*for every $i \in [N], k \in [n]$.*

*In both cases, the Transformer $\mathcal{N}$ has width 16 ($m = 6$, $h = k = 1$ and $q = 16$), depth*

$$O\left(n + \sqrt{N \log N} + \sqrt{\frac{N}{\log N}} \cdot \max\{\log C, \log R\}\right)$$

*and bit complexity bounded by*

$$O\left(\log d + \sqrt{\frac{N}{\log N}} \cdot \max\{\log C, \log R\}\right)$$

*where we denote $R := 8000 r^2 \delta^{-2} d n^5 N^6$.*

Theorem 4.1 shows that $\tilde{O}(d + n + \sqrt{N})$ parameters are enough to memorize $N$ sequence classification examples of length $n$ with token dimension $d$. Compared to the sequence-to-sequence mapping, there is $\sqrt{n}$ factor of saving in the last term.

## 4.2 LANGUAGE MODELING

We consider two language modeling tasks commonly used for pre-training Transformers: masked language modeling and autoregressive language modeling. We consider the memorization of all possible length $n$ sequences obtainable from the given corpus of length $T$. The input is embedded in the $d$-dimensional space while the output is mapped to one of $V$ tokens in the dictionary.

The memorization of masked language modeling requires $\tilde{O}(d + n + \sqrt{n^{m+1} m T})$ parameters (Theorem D.1) while the memorization of autoregressive language modeling requires $\tilde{O}(d + n + \sqrt{T})$ parameters (Theorem D.2). Compared to the autoregressive language modeling, the masked language modeling has the additional factor $n^m m$ that comes from memorizing all masking patterns separately and the factor $n$ that comes from memorizing all masked tokens instead of one next token. For more details on the settings and formal statement of the result, we refer to Appendix D.

## 5 EXPERIMENTS

### 5.1 EXPERIMENTAL SETUP

We complement our theory with experiments on real-world dataset. We train encoder-only Transformer models (Vaswani et al., 2017) on token classification task where each token is assigned a label as in Theorem 3.1 and sequence classification task where each sequence is assigned a label as in Theorem 4.1. We study the relationship between the memorized dataset size and the model size.

For token classification, we use 14,000 randomly selected examples among 14,041 training examples in the named entity recognition dataset from CoNLL-2003 (Tjong Kim Sang & De Meulder, 2003). For sequence classification, we use 50,000 randomly selected examples among 392,702 training examples in the MNLI dataset from GLUE benchmark (Wang et al., 2019).

We vary the model size through the embedding size $m$ while fixing the number of layers as $L = 6$. We fix the number of attention head as $h = 12$, the embedding to head size ratio as $m/k = h = 12$ and the feedforward to embedding size ratio as $q/m = 4$, as commonly done in practice. More details on experiments are in the Appendix F.

### 5.2 RESULTS

Figure 1 shows heatmaps of training errors as the dataset size and the model size vary. There is a clear trend that the training error is smaller (darker in color) for the smaller dataset size and the

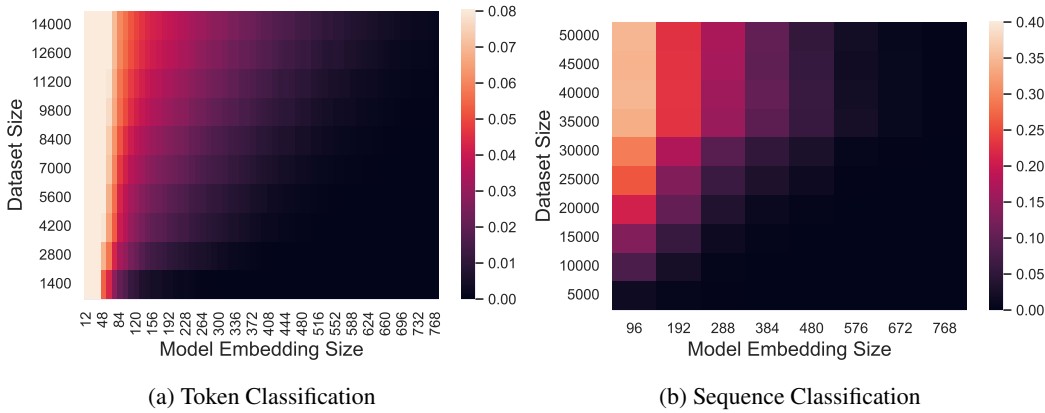

(a) Token Classification        (b) Sequence Classification

Figure 1: **Heatmaps of training errors.** We show the color-coded training errors as the dataset size and the model size vary. The dataset size is represented in the Y-axis while the model size is represented in the X-axis as the embedding size. The training error tends to get better for the smaller dataset size and the larger model size.

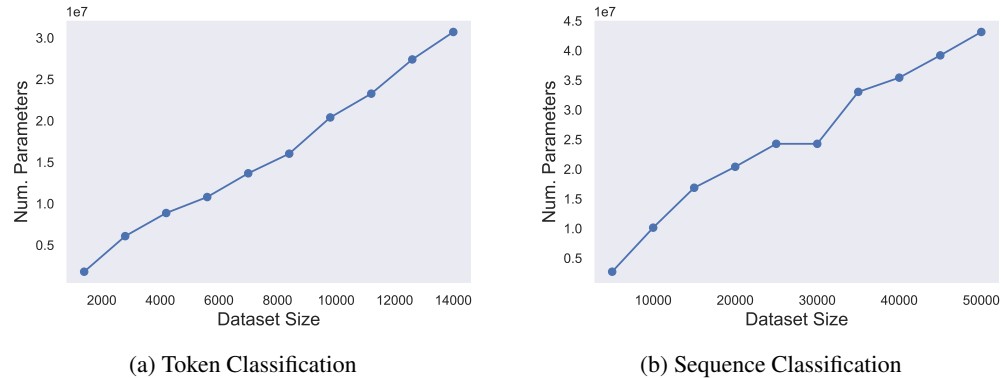

(a) Token Classification        (b) Sequence Classification

Figure 2: **The number of parameters required for memorization.** We show the number of parameters (Y-axis) of the smallest model that achieves less than 0.005 training error for each dataset size (X-axis). We observe a linear trend in the number of parameters with a subtle concavity as our theory predicts, but only at the low data regime.

larger model size. To see the clearer relationship between model size and the dataset size, we plot in Figure 2 the minimum model size that achieves less than 0.005 training error for each dataset size. In general, there is a linear increase in the model size as the memorized dataset size increases. We also observe a slight downward curvature (concavity) as our theory predicts, but only at the low dataset size. We conjecture that the linear trend may be due to the fixed depth and bit complexity during the experiments. See Remark 3.3 for the discussion of this bounded depth and bit complexity regime and see Appendix B for the formal result in these regimes. Indeed, Theorem B.1 and Theorem B.2 predicts linear dependence of the model size in the dataset size.

## 6 CONCLUSIONS

In this paper, we prove that Transformers are capable of memorizing $N$ length-$n$ sequence-to-sequence mappings with $\tilde{O}(d + n + \sqrt{nN})$ parameters. We extend our theory to analyze the memorization capacity of Transformers in other standard task settings. Our proof constructs a contextual mapping with $O(n)$ self-attention layers, which significantly improves the previously proposed selective-shifting-based contextual mapping in terms of parameter efficiency. Finally, we provide experimental results that verify our theory.

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

# A  PROOF OF THEOREM 3.1

Our proof of Theorem 3.1 consists of four stages as described in Section 3.3. We state and prove the main lemma for each stage in the following subsections. Then, we combine all stages in the end. The lemmas in each stage assumes permutation equivariance. We analyze how to circumvent permutation equivariance through the positional encoding in a separate subsection.

## A.1  STAGE 1: TOKENWISE PROJECTION

The main lemma for stage 1 is stated below.

**Lemma A.1.** *Let $N, d, n \in \mathbb{N}$ and $r \geq 1, 0 < \delta \leq 1$. Let $\boldsymbol{X}^{(1)}, \cdots, \boldsymbol{X}^{(N)} \in \mathbb{R}^{d \times n}$ be a set of $N$ input sequences that are distinct and tokenwise $(r, \delta)$-separated. Denote $r' = 2\lceil 2n^2 N^2 \sqrt{\pi d}\delta^{-1}\rceil \lceil r \rceil$.*

*Then, there exists a network $\mathcal{N}_1 : \mathbb{R}^{d \times n} \to \mathbb{R}^{1 \times n}$ consisting of the input embedding block and one tokenwise feedforward subblock with feedforward dimension $q = 1$ and bit complexity $\lceil \log(2rn^2 N^2 d \sqrt{\pi}\delta^{-1})\rceil$ such that $\mathcal{N}_1(\boldsymbol{X}^{(i)}), i \in [N]$ are non-negative and tokenwise $(2r', 2)$-separated. Moreover, for $i, j \in [N]$ and $k, l \in [n]$,*

$$\mathcal{N}_1(\boldsymbol{X}^{(i)})[1, k] = \mathcal{N}_1(\boldsymbol{X}^{(j)})[1, l] \text{ if and only if } \boldsymbol{X}^{(i)}[:, k] = \boldsymbol{X}^{(j)}[:, l].$$

*Proof.* Our proof defines a vector $\boldsymbol{u} \in \mathbb{R}^d$ such that sequences $\boldsymbol{x}^{(i)} = \boldsymbol{u}^T \boldsymbol{X}^{(i)} \in \mathbb{R}^{1 \times n}, i \in [N]$ are tokenwise $(r', 2)$-separated and satisfy that, for $i, j \in [N]$ and $k, l \in [n]$,

$$\boldsymbol{x}^{(i)}[1, k] = \boldsymbol{x}^{(j)}[1, l] \text{ if and only if } \boldsymbol{X}^{(i)}[:, k] = \boldsymbol{X}^{(j)}[:, l]$$

Then, we construct the network $\mathcal{N}_1$ with the required size and bit complexity that computes $\mathcal{N}_1(\boldsymbol{X}) = \boldsymbol{u}^T \boldsymbol{X} + r'\boldsymbol{1}_n^T$.

**Construction of $\boldsymbol{u}$.**  Recall the definition of the vocabulary $\mathcal{V} = \bigcup_{i \in [N]} \mathcal{V}^{(i)} = \{\boldsymbol{v} \in \mathbb{R}^d : \boldsymbol{v} = \boldsymbol{X}^{(i)}[:, k] \text{ for some } i \in [N], k \in [n]\}$. Note that $|\mathcal{V}| \leq nN$. We use Lemma E.1 on $\mathcal{V}$ to find $\tilde{\boldsymbol{u}}$ such that

$$\frac{1}{n^2 N^2}\sqrt{\frac{8}{\pi d}}\|\boldsymbol{v} - \boldsymbol{v}'\| \leq \frac{1}{|\mathcal{V}|^2}\sqrt{\frac{8}{\pi d}}\|\boldsymbol{v} - \boldsymbol{v}'\| \leq \left|\tilde{\boldsymbol{u}}^T(\boldsymbol{v} - \boldsymbol{v}')\right| \leq \|\boldsymbol{v} - \boldsymbol{v}'\|$$

for every $\boldsymbol{v}, \boldsymbol{v}' \in \mathcal{V}$.

Let $\hat{\boldsymbol{u}} \in \mathbb{R}^d$ be a vector with each coordinate being the first $\lceil \log(n^2 N^2 d\sqrt{\pi})\rceil$ bits of the corresponding coordinate of $\tilde{\boldsymbol{u}}$. We note that $\|\hat{\boldsymbol{u}} - \tilde{\boldsymbol{u}}\| \leq \frac{\sqrt{d}}{2^{\log(n^2 N^2 d\sqrt{\pi})}} = \frac{1}{n^2 N^2}\sqrt{\frac{1}{\pi d}}$.

Define $\boldsymbol{u} = S\hat{\boldsymbol{u}}$ with $S = \lceil 2n^2 N^2 \sqrt{\pi d}\delta^{-1}\rceil \geq 2$. We now check that $\boldsymbol{x}^{(i)} = \boldsymbol{u}^T \boldsymbol{X}^{(i)}, i \in [N]$ are tokenwise $(r', 2)$-separated with $r' = 2S\lceil r \rceil$. Let $i, j \in [N], k, l \in [n]$ with $\boldsymbol{X}^{(i)}[:, k] \neq \boldsymbol{X}^{(j)}[:, l]$ and $\boldsymbol{v}, \boldsymbol{v}' \in \mathcal{V}$ with $\boldsymbol{v} = \boldsymbol{X}^{(i)}[:, k], \boldsymbol{v}' = \boldsymbol{X}^{(j)}[:, l]$. Then, we have

$$\begin{aligned}
\left|\boldsymbol{x}^{(i)}[1, k]\right| &= \left|\boldsymbol{u}^T \boldsymbol{X}^{(i)}[:, k]\right| \\
&= S\left|\hat{\boldsymbol{u}}^T \boldsymbol{v}\right| \\
&\leq S\left(\left|\tilde{\boldsymbol{u}}^T \boldsymbol{v}\right| + \left|(\hat{\boldsymbol{u}} - \tilde{\boldsymbol{u}})^T \boldsymbol{v}\right|\right) \\
&\leq S\left(\|\boldsymbol{v}\| + \frac{1}{n^2 N^2}\sqrt{\frac{1}{\pi d}}\|\boldsymbol{v}\|\right) \\
&\leq 2S\|\boldsymbol{v}\| \\
&\leq 2Sr \leq r'.
\end{aligned}$$

We also have

$$
\begin{aligned}
\left| \boldsymbol{x}^{(i)}[1,k] - \boldsymbol{x}^{(j)}[1,l] \right| &= \left| \boldsymbol{u}^T (\boldsymbol{X}^{(i)}[:,k] - \boldsymbol{X}^{(j)}[:,l]) \right| \\
&= S \left| \hat{\boldsymbol{u}}^T (\boldsymbol{v} - \boldsymbol{v}') \right| \\
&\geq S \left( \left| \tilde{\boldsymbol{u}}^T (\boldsymbol{v} - \boldsymbol{v}') \right| - \left| (\hat{\boldsymbol{u}} - \tilde{\boldsymbol{u}})^T (\boldsymbol{v} - \boldsymbol{v}') \right| \right) \\
&\geq S \left( \frac{1}{n^2 N^2} \sqrt{\frac{8}{\pi d}} \|\boldsymbol{v} - \boldsymbol{v}'\| - \frac{1}{n^2 N^2} \sqrt{\frac{1}{\pi d}} \|\boldsymbol{v} - \boldsymbol{v}'\| \right) \\
&\geq S \frac{1}{n^2 N^2} \sqrt{\frac{1}{\pi d}} \|\boldsymbol{v} - \boldsymbol{v}'\| \\
&\geq S \frac{1}{n^2 N^2} \sqrt{\frac{1}{\pi d}} \delta \geq 2,
\end{aligned}
$$

which also implies

$$
\boldsymbol{x}^{(i)}[1,k] = \boldsymbol{x}^{(j)}[1,l] \text{ if and only if } \boldsymbol{X}^{(i)}[:,k] = \boldsymbol{X}^{(j)}[:,l].
$$

**Construction of $\mathcal{N}_1$.** We construct $\mathcal{N}_1$ as a composition of the input embedding block $\mathcal{E}_{in}$ : $\mathbb{R}^{n \times d} \to \mathbb{R}^{1 \times n}$ and a tokenwise feedforward block $\mathcal{F}^{(FF)} : \mathbb{R}^{1 \times n} \to \mathbb{R}^{1 \times n}$ with a skip-connection. We define $\mathcal{E}_{in}(\boldsymbol{X}) = \hat{\boldsymbol{u}}^T \boldsymbol{X}$ and

$$
\mathcal{F}^{(FF)}(\boldsymbol{z}) = \boldsymbol{z} + (S-1)\sigma_R(\boldsymbol{z} + 2\lceil r \rceil \mathbf{1}_n^T) + 2\lceil r \rceil \mathbf{1}_n^T.
$$

Then, we have

$$
\begin{aligned}
\mathcal{N}_1(\boldsymbol{X}) &= \mathcal{F}^{(FF)}(\mathcal{E}_{in}(\boldsymbol{X})) \\
&= \hat{\boldsymbol{u}}^T \boldsymbol{X} + (S-1)\sigma_R(\hat{\boldsymbol{u}}^T \boldsymbol{X} + 2\lceil r \rceil \mathbf{1}_n^T) + 2\lceil r \rceil \mathbf{1}_n^T \\
&= \hat{\boldsymbol{u}}^T \boldsymbol{X} + (S-1)(\hat{\boldsymbol{u}}^T \boldsymbol{X} + 2\lceil r \rceil \mathbf{1}_n^T) + 2\lceil r \rceil \mathbf{1}_n^T \\
&= S\hat{\boldsymbol{u}}^T \boldsymbol{X} + 2S\lceil r \rceil \mathbf{1}_n^T = \boldsymbol{u}^T \boldsymbol{X} + r' \mathbf{1}_n^T,
\end{aligned}
$$

where we removed the ReLU activation because $\hat{\boldsymbol{u}}^T \boldsymbol{X} + 2\lceil r \rceil \mathbf{1}_n^T$ have all positive values.

It is straightforward from the definition of $\mathcal{F}^{(FF)}$ that the feedforward dimension of the network is 1. Moreover, we can represent each coordinate of $\hat{\boldsymbol{u}}$ with $\lceil \log(n^2 N^2 d\sqrt{\pi}) \rceil$ bits, the weights in $\mathcal{F}^{(FF)}$ with $\lceil \log S \rceil = \lceil \log(2n^2 N^2 \sqrt{\pi d}\delta^{-1}) \rceil$ bits, and the biases in $\mathcal{F}^{(FF)}$ with $\lceil \log 2r \rceil$ bits. Thus, the bit complexity of the network $\mathcal{N}_1$ is $\lceil \log(2rn^2 N^2 d\sqrt{\pi}\delta^{-1}) \rceil$. $\qquad\square$

## A.2 Stage 2: Contextual mapping

Before proceeding on to the stage 2, we reindex the tokens of each output from Lemma A.1. Due to the permutation equivariance of the Transformer architecture, we may reorder tokens in any order as we want without loss of generality.

For each $i \in [N]$, we consider $\boldsymbol{x}^{(i)} \in \mathbb{R}^n$ as a vector and reindex as follows. Suppose that there are $n_i$ unique tokens in $\boldsymbol{X}^{(i)}$. Then, there are also $n_i$ unique values in $\boldsymbol{x}^{(i)}$. We assign indices for tokens so that the first $n_i$ tokens have $n_i$ unique values of $\boldsymbol{x}^{(i)}$ in descending order:

$$
\boldsymbol{x}^{(i)}[1] > \boldsymbol{x}^{(i)}[2] > \cdots > \boldsymbol{x}^{(i)}[n_i].
$$

Then, we index the remaining redundant tokens in descending order:

$$
\boldsymbol{x}^{(i)}[n_i + 1] \geq \boldsymbol{x}^{(i)}[n_i + 2] \geq \cdots \geq \boldsymbol{x}^{(i)}[n].
$$

We note that the resulting vector $\boldsymbol{x}^{(i)}$ is uniquely defined. That is, $\boldsymbol{X}^{(i)}$ corresponds to one and only one valid resulting vector $\boldsymbol{x}^{(i)}$.

We need the following definition in this section.

**Definition A.1.** Let $N, n \in \mathbb{N}$ and $r \geq 1, 0 < \delta \leq 1$. Let $\boldsymbol{x}^{(1)}, \cdots, \boldsymbol{x}^{(N)} \in \mathbb{R}^n$ be $N$ data instances. We say that $\boldsymbol{x}^{(1)}, \cdots, \boldsymbol{x}^{(N)}$ are $(r, \delta)$-separated if

- $\|\boldsymbol{x}^{(i)}\| \le r$ for all $i \in [N]$ and
- $\|\boldsymbol{x}^{(i)} - \boldsymbol{x}^{(j)}\| \ge \delta$ for all $i, j \in [N]$ with $i \ne j$.

We now state the main lemma for stage 2.

**Lemma A.2.** *Let $N, n, r' \in \mathbb{N}$. Let $\boldsymbol{x}^{(1)}, \cdots, \boldsymbol{x}^{(N)} \in \mathbb{R}^n$ be a set of $N$ input sequences that are non-negative and tokenwise $(2r', 2)$-separated. Denote $R' = 4\lceil 2N^2 \sqrt{\pi n} \rceil \lceil r' \rceil \lceil \sqrt{n} \rceil$.*

*Then, there exists a network $\mathcal{N}_2 : \mathbb{R}^{3 \times n} \to \mathbb{R}^{3 \times n}$ consisting of $2n$ Transformer blocks with the number of head $h = 1$, head size $k = 1$, feedforward dimension $q = 4$ and bit complexity $\lceil \log(2r'nN^2\sqrt{\pi}) \rceil$ such that*

$$\mathcal{N}_2 \left( \begin{bmatrix} \boldsymbol{x}^{(i)T} \\ \boldsymbol{0}_n^T \\ \boldsymbol{0}_n^T \end{bmatrix} \right) = \begin{bmatrix} \boldsymbol{0}_n^T \\ \boldsymbol{0}_n^T \\ z^{(i)} \boldsymbol{1}_n^T \end{bmatrix}$$

*where $z^{(i)} \in \mathbb{R}, i \in [N]$ are $(R' + 1, 2)$-separated.*

*Proof.* Our proof defines a vector $\boldsymbol{w} \in \mathbb{R}^n$ such that values $\tilde{z}^{(i)} = \boldsymbol{w}[1 : n_i]^T \boldsymbol{x}^{(i)}[1 : n_i] \in \mathbb{R}, i \in [N]$ are $(R', 4)$-separated where we denote the number of unique values in $\boldsymbol{x}^{(i)}$ as $n_i$. Then, we construct the network $\mathcal{N}_2$ with the required size and bit complexity that computes

$$\mathcal{N}_2 \left( \begin{bmatrix} \boldsymbol{x}^{(i)T} \\ \boldsymbol{0}_n^T \\ \boldsymbol{0}_n^T \end{bmatrix} \right) = \begin{bmatrix} \boldsymbol{0}_n^T \\ \boldsymbol{0}_n^T \\ z^{(i)} \boldsymbol{1}_n^T \end{bmatrix}$$

where $z^{(i)}$ approximates $\tilde{z}^{(i)}$ within 1 as

$$\left| z^{(i)} - \tilde{z}^{(i)} \right| \le 1.$$

Since we have

$$\left| z^{(i)} \right| \le \left| \tilde{z}^{(i)} \right| + \left| z^{(i)} - \tilde{z}^{(i)} \right| \le R' + 1$$

for $i \in [N]$ and

$$\left| z^{(i)} - z^{(j)} \right| \ge \left| \tilde{z}^{(i)} - \tilde{z}^{(j)} \right| - \left| z^{(i)} - \tilde{z}^{(i)} \right| - \left| z^{(j)} - \tilde{z}^{(j)} \right| \ge 4 - 1 - 1 = 2$$

for $i, j \in [N]$ with $i \ne j$, we conclude that $z^{(i)} \in \mathbb{R}, i \in [N]$ are $(R' + 1, 2)$-separated.

**Construction of $\boldsymbol{w}$.** For $i \in [N]$, we define $\tilde{\boldsymbol{x}}^{(i)} \in \mathbb{R}^n$ as a vector having the same values as $\boldsymbol{x}^{(i)}$ in the first $n_i$ coordinates and $0$ in the rest. We use Lemma E.1 on $\tilde{\boldsymbol{x}}^{(1)}, \tilde{\boldsymbol{x}}^{(2)}, \cdots, \tilde{\boldsymbol{x}}^{(N)}$ to find $\tilde{\boldsymbol{w}}$ such that

$$\frac{1}{N^2} \sqrt{\frac{8}{\pi n}} \|\tilde{\boldsymbol{x}}^{(i)} - \tilde{\boldsymbol{x}}^{(j)}\| \le \left| \tilde{\boldsymbol{w}}^T \left( \tilde{\boldsymbol{x}}^{(i)} - \tilde{\boldsymbol{x}}^{(j)} \right) \right| \le \|\tilde{\boldsymbol{x}}^{(i)} - \tilde{\boldsymbol{x}}^{(j)}\|$$

for every $i, j \in [N]$.

Let $\hat{\boldsymbol{w}} \in \mathbb{R}^d$ be a vector with each coordinate being the first $\lceil \log(nN^2\sqrt{\pi}) \rceil$ bits of the corresponding coordinate of $\tilde{\boldsymbol{w}}$. We note that $\|\hat{\boldsymbol{w}} - \tilde{\boldsymbol{w}}\| \le \frac{\sqrt{n}}{2^{\log(nN^2\sqrt{\pi})}} = \frac{1}{N^2} \sqrt{\frac{1}{\pi n}}$.

Define $\boldsymbol{w} = P\hat{\boldsymbol{w}}$ with $P = \lceil 2N^2\sqrt{\pi n} \rceil$. We now check that $\tilde{z}^{(i)} = \boldsymbol{w}[1:n_i]^T\boldsymbol{x}^{(i)}[1:n_i] = \boldsymbol{w}^T\tilde{\boldsymbol{x}}^{(i)}, i \in [N]$ are $(R', 4)$-separated. Let $i, j \in [N]$ with $i \neq j$. Then, we have

$$
\begin{aligned}
\left|\tilde{z}^{(i)}\right| &= \left|\boldsymbol{w}^T\tilde{\boldsymbol{x}}^{(i)}\right| \\
&= P\left|\hat{\boldsymbol{w}}^T\tilde{\boldsymbol{x}}^{(i)}\right| \\
&\leq P\left(\left|\tilde{\boldsymbol{w}}^T\tilde{\boldsymbol{x}}^{(i)}\right| + \left|(\hat{\boldsymbol{w}} - \tilde{\boldsymbol{w}})^T\tilde{\boldsymbol{x}}^{(i)}\right|\right) \\
&\leq P\left(\|\tilde{\boldsymbol{x}}^{(i)}\| + \frac{1}{N^2}\sqrt{\frac{1}{\pi n}}\|\tilde{\boldsymbol{x}}^{(i)}\|\right) \\
&\leq 2P\|\tilde{\boldsymbol{x}}^{(i)}\| \\
&= 2P\sqrt{\sum_{k=1}^{n}\left|\tilde{x}_k^{(i)}\right|^2} \\
&\leq 4Pr'\sqrt{n} \leq R'.
\end{aligned}
$$

and

$$
\begin{aligned}
\left|\tilde{z}^{(i)} - \tilde{z}^{(j)}\right| &= \left|\boldsymbol{w}^T(\tilde{\boldsymbol{x}}^{(i)} - \tilde{\boldsymbol{x}}^{(j)})\right| \\
&= P\left|\hat{\boldsymbol{w}}^T(\tilde{\boldsymbol{x}}^{(i)} - \tilde{\boldsymbol{x}}^{(j)})\right| \\
&\geq P\left(\left|\tilde{\boldsymbol{w}}^T(\tilde{\boldsymbol{x}}^{(i)} - \tilde{\boldsymbol{x}}^{(j)})\right| - \left|(\hat{\boldsymbol{w}} - \tilde{\boldsymbol{w}})^T(\tilde{\boldsymbol{x}}^{(i)} - \tilde{\boldsymbol{x}}^{(j)})\right|\right) \\
&\geq P\left(\frac{1}{N^2}\sqrt{\frac{8}{\pi n}}\|\tilde{\boldsymbol{x}}^{(i)} - \tilde{\boldsymbol{x}}^{(j)}\| - \frac{1}{N^2}\sqrt{\frac{1}{\pi n}}\|\tilde{\boldsymbol{x}}^{(i)} - \tilde{\boldsymbol{x}}^{(j)}\|\right) \\
&\geq P\frac{1}{N^2}\sqrt{\frac{1}{\pi n}}\|\tilde{\boldsymbol{x}}^{(i)} - \tilde{\boldsymbol{x}}^{(j)}\| \\
&\geq P\frac{1}{N^2}\sqrt{\frac{1}{\pi n}} \cdot 2 \geq 4.
\end{aligned}
$$

**Construction of $\mathcal{N}_2$.** We construct $\mathcal{N}_2 = \mathcal{F}_{2n} \circ \mathcal{F}_{2n-1} \circ \cdots \circ \mathcal{F}_1$ in $n$ steps. Each step $l \in [n]$ consists of 2 Transformer blocks $\mathcal{F}_{2l-1}, \mathcal{F}_{2l} : \mathbb{R}^{3 \times n} \to \mathbb{R}^{3 \times n}$.

First, we use Lemma E.2 to obtain a self-attention module $\tilde{\mathcal{F}}_{2l-1}^{(SA)} : \mathbb{R}^{1 \times n} \to \mathbb{R}^{1 \times n}$ that computes a vector with all coordinates holding $\frac{1}{2P\sqrt{n}}$-approximation $\hat{x}_{\max}$ of $x_{\max} = \max_{i \in [n]}\boldsymbol{x}[i]$ given $\boldsymbol{x} \in \mathbb{R}^n$. We extend $\tilde{\mathcal{F}}_{2l-1}^{(SA)}$ to define a valid self-attention subblock $\mathcal{F}_{2l-1}^{(SA)} : \mathbb{R}^{3 \times n} \to \mathbb{R}^{3 \times n}$ as

$$
\mathcal{F}_{2l-1}^{(SA)}\left(\begin{bmatrix} \boldsymbol{x}^T \\ \boldsymbol{0}_n^T \\ \boldsymbol{z}^T \end{bmatrix}\right) = \begin{bmatrix} \boldsymbol{x}^T \\ \boldsymbol{0}_n^T \\ \boldsymbol{z}^T \end{bmatrix} + \begin{bmatrix} \boldsymbol{0}_n^T \\ \tilde{\mathcal{F}}_{2l-1}^{(SA)}(\boldsymbol{x}^T) \\ \boldsymbol{0}_n^T \end{bmatrix} = \begin{bmatrix} \boldsymbol{x}^T \\ \hat{x}_{\max}\boldsymbol{1}_n^T \\ \boldsymbol{z}^T \end{bmatrix}
$$

for $\boldsymbol{x}, \boldsymbol{z} \in \mathbb{R}^n$. The subblock $\mathcal{F}_{2l-1}^{(SA)}$ finds the $\frac{1}{2P\sqrt{n}}$-approximate maximum token id among the first coordinate values and outputs in the second coordinate.

Next, we use Lemma E.3 to obtain a tokenwise feedforward module $\tilde{\mathcal{F}}_{2l-1}^{(FF)} : \mathbb{R}^{2 \times n} \to \mathbb{R}^{1 \times n}$ that outputs $r'$ for tokens with the same value in two coordinates. We define a tokenwise feedforward subblock $\mathcal{F}_{2l-1}^{(FF)} : \mathbb{R}^{3 \times n} \to \mathbb{R}^{3 \times n}$ by extending $\tilde{\mathcal{F}}_{2l-1}^{(FF)}$ so that, for $\boldsymbol{x}, \boldsymbol{y}, \boldsymbol{z} \in \mathbb{R}^n$,

$$
\mathcal{F}_{2l-1}^{(FF)}\left(\begin{bmatrix} \boldsymbol{x}^T \\ \boldsymbol{y}^T \\ \boldsymbol{z}^T \end{bmatrix}\right) = \begin{bmatrix} \boldsymbol{x}^T \\ \boldsymbol{y}^T \\ \boldsymbol{z}^T \end{bmatrix} - \begin{bmatrix} 2\tilde{\mathcal{F}}_{2l-1}^{(FF)}([\boldsymbol{x}, \boldsymbol{y}]^T) \\ \boldsymbol{0}_n^T \\ \boldsymbol{0}_n^T \end{bmatrix} = \begin{bmatrix} \tilde{\boldsymbol{x}}^T \\ \boldsymbol{y}^T \\ \boldsymbol{z}^T \end{bmatrix},
$$

where

$$
\tilde{\boldsymbol{x}}[i] = \begin{cases} \boldsymbol{x}[i] - 2r' & \text{if } |\boldsymbol{x}[i] - \boldsymbol{y}[i]| < \frac{1}{2} \\ \boldsymbol{x}[i] & \text{if } |\boldsymbol{x}[i] - \boldsymbol{y}[i]| > 1 \end{cases}.
$$

The subblock $\mathcal{F}_{2l-1}^{(FF)}$ compares the first two rows of the input and subtracts $2r'$ from the first row if two values are not separated. Since $\boldsymbol{x}[i]$ is bounded above by $2r'$, the subtracted entries become negative.

Since we do not need the self-attention subblock from $\mathcal{F}_{2l}$, we set all weights of $\mathcal{F}_{2l}^{(SA)}$ to zero. We define $\mathcal{F}_{2l}^{(FF)} : \mathbb{R}^{3\times n} \to \mathbb{R}^{3\times n}$ as

$$
\begin{aligned}
\mathcal{F}_{2l}^{(FF)}\left(\begin{bmatrix} \boldsymbol{x}^T \\ \boldsymbol{y}^T \\ \boldsymbol{z}^T \end{bmatrix}\right) &= \begin{bmatrix} \boldsymbol{x}^T \\ \boldsymbol{y}^T \\ \boldsymbol{z}^T \end{bmatrix} + \begin{bmatrix} 1 & 0 \\ 0 & -1/P \\ 0 & \hat{w}_k \end{bmatrix} \sigma_R\left(\begin{bmatrix} -\mathbf{e}_1^T \\ P\mathbf{e}_2^T \end{bmatrix}\begin{bmatrix} \boldsymbol{x}^T \\ \boldsymbol{y}^T \\ \boldsymbol{z}^T \end{bmatrix}\right) \\
&= \begin{bmatrix} \boldsymbol{x}^T \\ \boldsymbol{y}^T \\ \boldsymbol{z}^T \end{bmatrix} + \begin{bmatrix} 1 & 0 \\ 0 & -1/P \\ 0 & \hat{w}_k \end{bmatrix} \sigma_R\left(\begin{bmatrix} -\boldsymbol{x}^T \\ P\boldsymbol{y}^T \end{bmatrix}\right) \\
&= \begin{bmatrix} \boldsymbol{x}^T + \sigma_R(-\boldsymbol{x}^T) \\ \boldsymbol{y}^T - \sigma_R(\boldsymbol{y}^T) \\ \boldsymbol{z}^T + \hat{w}_k P\sigma_R(\boldsymbol{y}^T) \end{bmatrix} \\
&= \begin{bmatrix} \sigma_R(\boldsymbol{x}^T) \\ -\sigma_R(-\boldsymbol{y}^T) \\ \boldsymbol{z}^T + w_k\sigma_R(\boldsymbol{y}^T) \end{bmatrix},
\end{aligned}
$$

where $P > 0$ and $w_k = \hat{w}_k P$ are defined earlier. When $\boldsymbol{y} \geq \mathbf{0}_n$, we can further simplify as

$$
\mathcal{F}_{2l}^{(FF)}\left(\begin{bmatrix} \boldsymbol{x}^T \\ \boldsymbol{y}^T \\ \boldsymbol{z}^T \end{bmatrix}\right) = \begin{bmatrix} \sigma_R(\boldsymbol{x}^T) \\ \mathbf{0}_n^T \\ \boldsymbol{z}^T + w_k\boldsymbol{y}^T \end{bmatrix}.
$$

Let $\boldsymbol{Z}^{(i,l)} = \mathcal{F}_{2l} \circ \mathcal{F}_{2l-1} \circ \cdots \circ \mathcal{F}_1\left(\begin{bmatrix} \boldsymbol{x}^{(i)T} \\ \mathbf{0}_n^T \\ \mathbf{0}_n^T \end{bmatrix}\right) \in \mathbb{R}^{3\times n}$ be the output of the $l$-th step when the input is $\boldsymbol{x}^{(i)}$ for $l = 0, \cdots, n$. We show inductively that

$$
\begin{aligned}
\boldsymbol{Z}^{(i,l)}[1,k] &= \begin{cases} \boldsymbol{x}^{(i)}[k] & \text{if } l < n_i \text{ and } \boldsymbol{x}^{(i)}[k] < \boldsymbol{x}^{(i)}[l] \\ 0 & \text{otherwise} \end{cases}, \\
\boldsymbol{Z}^{(i,l)}[2,k] &= 0, \\
\boldsymbol{Z}^{(i,l)}[3,k] &= \boldsymbol{w}[1:\min\{l,n_i\}]^T\hat{\boldsymbol{x}}^{(i)}[1:\min\{l,n_i\}] = \sum_{j=1}^{\min\{l,n_i\}} \boldsymbol{w}[j]\hat{\boldsymbol{x}}^{(i)}[j]
\end{aligned} \tag{1}
$$

for $i \in [N], l, k \in [n]$, where $\hat{\boldsymbol{x}}^{(i)}[j]$ is $\frac{1}{2P\sqrt{n}}$-approximation of $\boldsymbol{x}^{(i)}[j]$.

For $l = 0$, the conditions 1 hold for the input $\begin{bmatrix} \boldsymbol{x}^{(i)T} \\ \mathbf{0}_n^T \\ \mathbf{0}_n^T \end{bmatrix}$. Suppose that the conditions 1 hold for $l = \tilde{l} - 1$. When $\tilde{l} > n_i$, the induction hypothesis implies that $\boldsymbol{Z}^{(i,\tilde{l}-1)}[1,:] = \mathbf{0}_n^T$. Thus, we obtain

the output as

$$\mathcal{F}_{2\tilde{l}}^{(FF)} \circ \mathcal{F}_{2\tilde{l}-1}^{(FF)} \circ \mathcal{F}_{2\tilde{l}-1}^{(SA)} \left( \boldsymbol{Z}^{(i,\tilde{l}-1)} \right) = \mathcal{F}_{2\tilde{l}}^{(FF)} \circ \mathcal{F}_{2\tilde{l}-1}^{(FF)} \circ \mathcal{F}_{2\tilde{l}-1}^{(SA)} \left( \begin{bmatrix} \boldsymbol{0}_n^T \\ \boldsymbol{0}_n^T \\ \boldsymbol{Z}^{(i,\tilde{l}-1)}[3,:] \end{bmatrix} \right)$$

$$= \mathcal{F}_{2\tilde{l}}^{(FF)} \circ \mathcal{F}_{2\tilde{l}-1}^{(FF)} \left( \begin{bmatrix} \boldsymbol{0}_n^T \\ \boldsymbol{0}_n^T \\ \boldsymbol{Z}^{(i,\tilde{l}-1)}[3,:] \end{bmatrix} \right)$$

$$= \mathcal{F}_{2\tilde{l}}^{(FF)} \left( \begin{bmatrix} -2r'\boldsymbol{1}_n^T \\ \boldsymbol{0}_n^T \\ \boldsymbol{Z}^{(i,\tilde{l}-1)}[3,:] \end{bmatrix} \right)$$

$$= \begin{bmatrix} \boldsymbol{0}_n^T \\ \boldsymbol{0}_n^T \\ \boldsymbol{Z}^{(i,\tilde{l}-1)}[3,:] \end{bmatrix}.$$

When $\tilde{l} \leq n_i$, the induction hypothesis implies that $\boldsymbol{Z}^{(i,\tilde{l}-1)}[1,:]$ has non-zero entries among which the largest value is $\boldsymbol{x}^{(i)}[\tilde{l}]$. Then, it follows that

$$\mathcal{F}_{2\tilde{l}}^{(FF)} \circ \mathcal{F}_{2\tilde{l}-1}^{(FF)} \circ \mathcal{F}_{2\tilde{l}-1}^{(SA)} \left( \boldsymbol{Z}^{(i,\tilde{l}-1)} \right) = \mathcal{F}_{2\tilde{l}}^{(FF)} \circ \mathcal{F}_{2\tilde{l}-1}^{(FF)} \circ \mathcal{F}_{2\tilde{l}-1}^{(SA)} \left( \begin{bmatrix} \boldsymbol{Z}^{(i,\tilde{l}-1)}[1,:] \\ \boldsymbol{Z}^{(i,\tilde{l}-1)}[2,:] \\ \boldsymbol{Z}^{(i,\tilde{l}-1)}[3,:] \end{bmatrix} \right)$$

$$= \mathcal{F}_{2\tilde{l}}^{(FF)} \circ \mathcal{F}_{2\tilde{l}-1}^{(FF)} \circ \mathcal{F}_{2\tilde{l}-1}^{(SA)} \left( \begin{bmatrix} \boldsymbol{Z}^{(i,\tilde{l}-1)}[1,:] \\ \boldsymbol{0}_n^T \\ \boldsymbol{Z}^{(i,\tilde{l}-1)}[3,:] \end{bmatrix} \right)$$

$$= \mathcal{F}_{2\tilde{l}}^{(FF)} \circ \mathcal{F}_{2\tilde{l}-1}^{(FF)} \left( \begin{bmatrix} \boldsymbol{Z}^{(i,\tilde{l}-1)}[1,:] \\ \hat{\boldsymbol{x}}^{(i)}[\tilde{l}]\boldsymbol{1}_n^T \\ \boldsymbol{Z}^{(i,\tilde{l}-1)}[3,:] \end{bmatrix} \right)$$

$$= \mathcal{F}_{2\tilde{l}}^{(FF)} \left( \begin{bmatrix} \tilde{\boldsymbol{x}}^{(i,\tilde{l}-1)T} \\ \hat{\boldsymbol{x}}^{(i)}[\tilde{l}]\boldsymbol{1}_n^T \\ \boldsymbol{Z}^{(i,\tilde{l}-1)}[3,:] \end{bmatrix} \right)$$

$$= \begin{bmatrix} \sigma_R \left( \tilde{\boldsymbol{x}}^{(i,\tilde{l}-1)T} \right) \\ \boldsymbol{0}_n^T \\ \boldsymbol{Z}^{(i,\tilde{l}-1)}[3,:] + \boldsymbol{w}[\tilde{l}]\hat{\boldsymbol{x}}^{(i)}[\tilde{l}]\boldsymbol{1}_n^T \end{bmatrix}$$

where $\hat{\boldsymbol{x}}^{(i)}[\tilde{l}]$ is $\frac{1}{2P\sqrt{n}}$-approximation of $\boldsymbol{x}^{(i)}[\tilde{l}]$. Here, $\tilde{\boldsymbol{x}}^{(i,\tilde{l}-1)}$ denotes

$$\tilde{\boldsymbol{x}}^{(i,\tilde{l}-1)}[k] = \begin{cases} \boldsymbol{Z}^{(i,\tilde{l}-1)}[1,k] - 2r' & \text{if } |\boldsymbol{Z}^{(i,\tilde{l}-1)}[1,k] - \hat{\boldsymbol{x}}^{(i)}[\tilde{l}]| < \frac{1}{2} \\ \boldsymbol{Z}^{(i,\tilde{l}-1)}[1,k] & \text{if } |\boldsymbol{Z}^{(i,\tilde{l}-1)}[1,k] - \hat{\boldsymbol{x}}^{(i)}[\tilde{l}]| > 1 \end{cases}$$

so

$$\sigma_R(\tilde{\boldsymbol{x}}^{(i,\tilde{l}-1)}[k]) = \begin{cases} 0 & \text{if } |\boldsymbol{Z}^{(i,\tilde{l}-1)}[1,k] - \hat{\boldsymbol{x}}^{(i)}[\tilde{l}]| < \frac{1}{2} \\ \boldsymbol{Z}^{(i,\tilde{l}-1)}[1,k] & \text{if } |\boldsymbol{Z}^{(i,\tilde{l}-1)}[1,k] - \hat{\boldsymbol{x}}^{(i)}[\tilde{l}]| > 1 \end{cases}.$$

Since distinct tokens are separated by 2, $\frac{1}{2P\sqrt{n}}$-approximations of them are separated by 1. On the other hand, since $\frac{1}{2P\sqrt{n}} < \frac{1}{2}$, the $\frac{1}{2P\sqrt{n}}$-approximation of the maximum element stays closer than $\frac{1}{2}$. Consequently, $\sigma_R(\tilde{\boldsymbol{x}}^{(i,\tilde{l})T})$ is the same as $\boldsymbol{Z}^{(i,\tilde{l})}[1,:]$. Thus, the conditions 1 hold and we conclude our induction proof.

In the end of $n$ steps, the output is $\boldsymbol{Z}^{(i,n)} = \begin{bmatrix} \boldsymbol{0}_n^T \\ \boldsymbol{0}_n^T \\ z^{(i)}\boldsymbol{1}_n^T \end{bmatrix}$ with $z^{(i)} = \boldsymbol{w}^T \hat{\boldsymbol{x}}^{(i)}$ where each entry of $\hat{\boldsymbol{x}}^{(i)}$ $\frac{1}{2P\sqrt{n}}$-approximates the corresponding entries of $\tilde{\boldsymbol{x}}^{(i)}$. We check that $z^{(i)}$ approximates $\tilde{z}^{(i)}$ within

1 as

$$\begin{aligned}
\left| z^{(i)} - \tilde{z}^{(i)} \right| &\leq \left| \boldsymbol{w}^T \hat{\boldsymbol{x}}^{(i)} - \boldsymbol{w}^T \tilde{\boldsymbol{x}}^{(i)} \right| \\
&\leq \|\boldsymbol{w}\| \|\hat{\boldsymbol{x}}^{(i)} - \tilde{\boldsymbol{x}}^{(i)}\| \\
&\leq P \|\hat{\boldsymbol{w}}\| \frac{\sqrt{n}}{2P\sqrt{n}} \\
&\leq \frac{1}{2} \left( \|\tilde{\boldsymbol{w}}\| + \|\hat{\boldsymbol{w}} - \tilde{\boldsymbol{w}}\| \right) \\
&\leq \frac{1}{2} \left( 1 + \frac{1}{N^2} \sqrt{\frac{1}{\pi n}} \right) \leq 1.
\end{aligned}$$

Our construction of $\mathcal{N}_2$ involves $2n$ Transformer blocks. From Lemma E.2, the subblock $\mathcal{F}_{2l-1}^{(SA)}$ uses 1 head ($h = 1$) with head size 1 ($k = 1$) and bit complexity $\lceil \log \left( \log(8n^{3/2} r' P) \right) \rceil = \lceil \log \left( \log(16n^2 N^2 r' \sqrt{\pi}) \right) \rceil$. From Lemma E.3, the subblock $\mathcal{F}_{2l-1}^{(FF)}$ uses feedforward dimension 4 with bit complexity $\lceil \log 2r' \rceil$. The subblock $\mathcal{F}_{2l}^{(FF)}$ uses feedforward dimension 2. All weights in $\mathcal{F}_{2l}^{(FF)}$ are either $0, \pm 1, P, -1/P$ or $\hat{w}_k$. Since we represent each coordinate of $\hat{w}_k$ using $\lceil \log(nN^2 \sqrt{\pi}) \rceil$ bits, the bit complexity of $\mathcal{F}_{2l}^{(FF)}$ is $\max\{\lceil \log P \rceil, \lceil \log(nN^2 \sqrt{\pi}) \rceil\} = \lceil \log(2nN^2 \sqrt{\pi}) \rceil$. Thus, the network $\tilde{\mathcal{N}}_2$ has 1 head ($h = 1$) with the head size 1 ($k = 1$), the feedforward dimension 4 and the bit complexity $\lceil \log(2r' nN^2 \sqrt{\pi}) \rceil$. $\qquad\square$

### A.3 Stage 3: String lookup

In stages 3 and 4, we map each token $\boldsymbol{X}^{(i)}[:, k]$ to the corresponding label using both the token id $\boldsymbol{x}^{(i)}[k]$ from stage 1 and the sequence id $z^{(i)}$ from stage 2. Now that the token id $\boldsymbol{x}^{(i)}[k]$ and the sequence id $z^{(i)}$ hold enough information to identify the label, we process each token independently using tokenwise feedforward blocks from now on. We first combine two ids into a single "contextual token" id and map to the corresponding token label in the last two stages. We adopt stages 2 and 3 in Vardi et al. (2022) to our architecture that involves skip-connections.

We can extend stage 2 by using extra dimension to pass $\boldsymbol{x}^{(i)}$ from stage 1 without any additional parameter. Then, after stage 2, the $k$-th token in the $i$-th sequence contains $z^{(i)}$ and $\boldsymbol{x}^{(i)}[k]$, which are enough to identify the corresponding label $\boldsymbol{y}^{(i)}[k]$. We note that $0 \leq \boldsymbol{x}^{(i)}[k] \leq 2r'$ and $|z^{(i)}| \leq R' + 1$ with $r', R' > 6$. We define $\boldsymbol{a}^{(i)}[k] = (\lfloor z^{(i)} \rfloor + R' + 1)(2r' + 1) + \lfloor \boldsymbol{x}^{(i)}[k] \rfloor + 1$ to be the unique integer id for each token in each sequence. Then, we have

- $1 \leq \boldsymbol{a}^{(i)}[k] < (2R' + 3)(2r' + 1) < 9r'R'$ for $i \in [N]$ and
- $\left| \boldsymbol{a}^{(i)}[k] - \boldsymbol{a}^{(j)}[l] \right| \geq 2$ for $i, j \in [N], k, l \in [n]$ with $\mathcal{V}^{(i)} \neq \mathcal{V}^{(j)}$ or $\boldsymbol{X}^{(i)}[:, k] \neq \boldsymbol{X}^{(j)}[:, l]$.

We denote $R = 9r'R'$. Now, our goal is to map $\boldsymbol{a}^{(i)}[k] \in [R]$ to $\boldsymbol{y}^{(i)}[k] \in [C]$ using tokenwise feedforward subblocks. We denote the number of distinct $\boldsymbol{a}^{(i)}[k]$'s as $N' \leq nN$. We reindex each of unique tokens and the corresponding label as $\tilde{a}^{(i)}$ and $\tilde{y}^{(i)}$ for $i \in [N']$, respectively. Without loss of generality, we suppppose that $1 \leq \tilde{a}^{(1)} < \cdots < \tilde{a}^{(N')} < R$.

Then, we partition $N'$ contextual token ids into $A$ groups of $B$ ids. For each group $g \in [A]$, we construct two strings $u_g$ and $w_g$. The binary string $u_g$ is a concatenation of $B$ ids in the group $g$. Each id is represented as an integer of $\rho = \lceil \log R \rceil$ bits. The binary string $w_g$ is a concatenation of $B$ labels corresponding to $B$ ids in the group $g$. Each label is represented as an integer of $\gamma = \lceil \log C \rceil$ bits. Thus, $u_g$ and $w_g$ are strings of length $\rho B$ and $\gamma B$, respectively.

We now state the main lemma for stage 3.

**Lemma A.3.** *Let $N', R \in \mathbb{N}$ and $1 \leq \tilde{a}^{(1)} < \cdots < \tilde{a}^{(N')} < R$ to be distinct integer ids that identify each token in each sequence. We suppose that $\left| \tilde{a}^{(i)} - \tilde{a}^{(j)} \right| \geq 2$ for $i, j \in [N']$ with $i \neq j$. Let $A, B, b \in \mathbb{N}$ with $A < N'$ and $B = \lceil \frac{N'}{A} \rceil$. Let $w_1, \cdots, w_A \in \mathbb{N}$ with the number of bits in their binary representation at most $b$.*

*Then, there exists a network* $\mathcal{N}_3 : \mathbb{R}^2 \to \mathbb{R}^2$ *consisting of A feedforward blocks with skip-connections every 2 layers, feedforward dimension* $q = 4$ *and bit complexity* $b + \lceil \log(2R + 1) \rceil$ *such that*

$$\mathcal{N}_3\left(\tilde{a}^{(i)}, 0\right) = \left(\tilde{a}^{(i)}, w_{\lceil \frac{i}{B} \rceil}\right).$$

*Proof.* We construct $\mathcal{N}_3 = \mathcal{F}_A \circ \mathcal{F}_{A-1} \circ \cdots \circ \mathcal{F}_1$ as a composition of $A$ feedforward blocks $\mathcal{F}_l : \mathbb{R}^2 \to \mathbb{R}^2, l \in [A]$ where each feedforward block contains a single hidden layer and a skip-connection.

Let $l \in [A]$. We use Lemma E.4 to define $\tilde{\mathcal{F}}_l : \mathbb{R}^2 \to \mathbb{R}^1$ such that $\tilde{\mathcal{F}}_l(x) = w_l$ if $x \in \left[\tilde{a}^{((l-1)\cdot B+1)}, \tilde{a}^{(l\cdot B)}\right]$ and $\tilde{\mathcal{F}}_i(x) = 0$ if $x \notin \left[\tilde{a}^{((l-1)\cdot B+1)} - \frac{1}{2}, \tilde{a}^{(l\cdot B)} + \frac{1}{2}\right]$ where we regard $l \cdot B > N'$ to be $N'$. We extend $\tilde{\mathcal{F}}_l$ to define a valid feedforward block $\mathcal{F}_l$ as

$$\mathcal{F}_l(x, y) = (x, y) + \left(0, \tilde{\mathcal{F}}_i(x)\right) = \left(x, y + \tilde{\mathcal{F}}_i(x)\right).$$

Throughout the computation of $\mathcal{N}_3$, the first coordinate is the same across all blocks. For $i \in [N']$, $\tilde{a}^{(i)}$ only activates one of $\tilde{\mathcal{F}}_l, l \in [A]$. In particular, $\tilde{\mathcal{F}}_l(\tilde{a}^{(i)}) = w_l$ if $l = \lceil \frac{i}{B} \rceil$ and $\tilde{\mathcal{F}}_l(\tilde{a}^{(i)}) = 0$ otherwise. Thus, we get

$$\mathcal{N}_3\left(\tilde{a}^{(i)}, 0\right) = \left(\tilde{a}^{(i)}, w_{\lceil \frac{i}{B} \rceil}\right).$$

Our construction involves $A$ feedforward blocks with skip-connections every 2 layers. From Lemma E.4, feedforward dimension $q = 4$ and the bit complexity is $b + \lceil \log(2R + 1) \rceil$. $\square$

**Remark A.4.** *We can extend the construction from Lemma A.3 to output multiple strings corresponding to the same range without additional feedforward dimension. In particular, the first layer in the construction of Lemma E.4 does not depend on* $w$. *Thus, we can reuse this four units with different output weights to output additional strings corresponding to the same range. In stage 4, we need 2 strings for each range.*

## A.4 STAGE 4: BIT EXTRACTION

We first define $\text{BIN}_{i:j}(n)$ to be the substring of $n$ with bits in places from $i$ to $j$ inclusive for $n, i, j \in \mathbb{N}$ with $i \leq j$. We now state the main lemma for stage 4.

**Lemma A.5.** *Let* $B, \rho, \gamma \in \mathbb{N}$ *and* $u, w \in \mathbb{N}$. *Suppose that the number of bits in binary representation of* $u$ *and* $w$ *are* $\rho B$ *and* $\gamma B$, *respectively. We assume that*

$$\left| \text{BIN}_{\rho\cdot(i-1)+1:\rho\cdot i}(u) - \text{BIN}_{\rho\cdot(j-1)+1:\rho\cdot j}(u) \right| \geq 2$$

*for* $i, j \in [B]$ *with* $i \neq j$.

*Then, there exists a network* $\mathcal{N}_4 : \mathbb{R}^3 \to \mathbb{R}$ *consisting of an output embedding block and* $(\max\{\rho, \gamma\} + 2)B + 2$ *feedforward blocks with skip-connections every 2 layers, feedforward dimension* $q = 16$ *and bit complexity* $2\max\{\rho, \gamma\}B$ *such that*

$$\mathcal{N}_4(x, u, w) = \text{BIN}_{\rho\cdot(i-1)+1:\rho\cdot i}(w),$$

*if there exist* $i \in [B]$ *such that* $x = \text{BIN}_{\rho\cdot(i-1)+1:\rho\cdot i}(u)$.

*Proof.* We construct $\mathcal{N}_4 = \mathcal{F}_{post} \circ \mathcal{F}_B \circ \mathcal{F}_{B-1} \circ \cdots \circ \mathcal{F}_1 \circ \mathcal{F}_{pre}$ in $B$ steps with pre-processing $\mathcal{F}_{pre} : \mathbb{R}^3 \to \mathbb{R}^8$ and post-processing $\mathcal{F}_{post} : \mathbb{R}^8 \to \mathbb{R}$. We define the pre-processing network as

$$\mathcal{F}_{pre}(x, u, w) = \left(x, \frac{u}{2^{\rho B}} + \frac{1}{2^{\rho B+1}}, \frac{u}{2^{\rho B}} + \frac{1}{2^{\rho B+2}}, 0, \frac{w}{2^{\gamma B}} + \frac{1}{2^{\gamma B+1}}, \frac{w}{2^{\gamma B}} + \frac{1}{2^{\gamma B+2}}, 0, 0\right)$$

and the post-processing network as

$$\mathcal{F}_{post}(z_1, z_2, \cdots, z_8) = z_8.$$

In each step $l \in [B]$, $\mathcal{F}_l : \mathbb{R}^8 \to \mathbb{R}^8$ first implements two sub-networks $\mathcal{F}_l^u : \mathbb{R}^3 \to \mathbb{R}^3$ and $\mathcal{F}_l^w : \mathbb{R}^3 \to \mathbb{R}^3$ in parallel on the middle 6 coordinates. Two sub-networks uses Lemma E.5 to compute

$$
\mathcal{F}_l^u \left( \begin{bmatrix} \phi^{(\rho(l-1))}\left(\frac{u}{2^\rho B} + \frac{1}{2^\rho B+1}\right) \\ \phi^{(\rho(l-1))}\left(\frac{u}{2^\rho B} + \frac{1}{2^\rho B+2}\right) \\ 0 \end{bmatrix} \right) = \begin{bmatrix} \phi^{(\rho l)}\left(\frac{u}{2^\rho B} + \frac{1}{2^\rho B+1}\right) \\ \phi^{(\rho l)}\left(\frac{u}{2^\rho B} + \frac{1}{2^\rho B+2}\right) \\ \mathrm{BIN}_{\rho(l-1)+1:\rho l}(u) \end{bmatrix}
$$

and

$$
\mathcal{F}_l^w \left( \begin{bmatrix} \phi^{(\gamma(l-1))}\left(\frac{w}{2^\gamma B} + \frac{1}{2^\gamma B+1}\right) \\ \phi^{(\gamma(l-1))}\left(\frac{w}{2^\gamma B} + \frac{1}{2^\gamma B+2}\right) \\ 0 \end{bmatrix} \right) = \begin{bmatrix} \phi^{(\gamma l)}\left(\frac{w}{2^\gamma B} + \frac{1}{2^\gamma B+1}\right) \\ \phi^{(\gamma l)}\left(\frac{w}{2^\gamma B} + \frac{1}{2^\gamma B+2}\right) \\ \mathrm{BIN}_{\gamma(l-1)+1:\gamma l}(w) \end{bmatrix} .
$$

Then, $\mathcal{F}_l$ combines the result using two additional feedforward blocks as

$$
\begin{bmatrix} x \\ \phi^{(\rho l)}\left(\frac{u}{2^\rho B} + \frac{1}{2^\rho B+1}\right) \\ \phi^{(\rho l)}\left(\frac{u}{2^\rho B} + \frac{1}{2^\rho B+2}\right) \\ \mathrm{BIN}_{\rho(l-1)+1:\rho l}(u) \\ \phi^{(\gamma l)}\left(\frac{w}{2^\gamma B} + \frac{1}{2^\gamma B+1}\right) \\ \phi^{(\gamma l)}\left(\frac{w}{2^\gamma B} + \frac{1}{2^\gamma B+2}\right) \\ \mathrm{BIN}_{\gamma(l-1)+1:\gamma l}(w) \\ 0 \end{bmatrix} \xrightarrow{\mathcal{F}_l^1} \begin{bmatrix} x \\ \phi^{(\rho l)}\left(\frac{u}{2^\rho B} + \frac{1}{2^\rho B+1}\right) \\ \phi^{(\rho l)}\left(\frac{u}{2^\rho B} + \frac{1}{2^\rho B+2}\right) \\ c_l = \tilde{\mathcal{F}}_l^1(x, \mathrm{BIN}_{\rho(l-1)+1:\rho l}(u)) \\ \phi^{(\gamma l)}\left(\frac{w}{2^\gamma B} + \frac{1}{2^\gamma B+1}\right) \\ \phi^{(\gamma l)}\left(\frac{w}{2^\gamma B} + \frac{1}{2^\gamma B+2}\right) \\ \mathrm{BIN}_{\gamma(l-1)+1:\gamma l}(w) \\ 0 \end{bmatrix}
$$

$$
\xrightarrow{\mathcal{F}_l^2} \begin{bmatrix} x \\ \phi^{(\rho l)}\left(\frac{u}{2^\rho B} + \frac{1}{2^\rho B+1}\right) \\ \phi^{(\rho l)}\left(\frac{u}{2^\rho B} + \frac{1}{2^\rho B+2}\right) \\ 0 \\ \phi^{(\gamma l)}\left(\frac{w}{2^\gamma B} + \frac{1}{2^\gamma B+1}\right) \\ \phi^{(\gamma l)}\left(\frac{w}{2^\gamma B} + \frac{1}{2^\gamma B+2}\right) \\ 0 \\ \tilde{\mathcal{F}}_l^2(c_l, \mathrm{BIN}_{\gamma(l-1)+1:\gamma l}(w)) \end{bmatrix}
$$

where $\tilde{\mathcal{F}}_l^1$ uses Lemma E.3 to compute

$$
\tilde{\mathcal{F}}_l^1(x, y) = \begin{cases} 2^\gamma & \text{if } |x - y| < \frac{1}{2} \\ 0 & \text{if } |x - y| > 1 \end{cases}
$$

and $\tilde{\mathcal{F}}_l^2$ computes, for $0 \le y \le 2^\gamma$

$$
\tilde{\mathcal{F}}_l^2(x, y) = \sigma_R(x - 2^\gamma + y) = \begin{cases} y & \text{if } x = 2^\gamma \\ 0 & \text{if } x = 0 \end{cases} .
$$

The resulting value in the last coordinate is $\mathrm{BIN}_{\gamma(l-1)+1:\gamma l}(w)$ if $|x - \mathrm{BIN}_{\rho(l-1)+1:\rho l}(u)| < \frac{1}{2}$ and 0 if $|x - \mathrm{BIN}_{\rho(l-1)+1:\rho l}(u)| > 1$. Therefore, the last coordinate throughout each step of $\mathcal{N}_4$ keeps the value 0 until it finds $x = \mathrm{BIN}_{\rho(l-1)+1:\rho l}(u)$. When such $l$ is found, the value is updated to $\mathrm{BIN}_{\gamma(l-1)+1:\gamma l}(w)$ as the requirement.

Finally, the parallel implementation of $\mathcal{F}_l^u$ and $\mathcal{F}_l^w$ requires $\max\{\rho, \gamma\}$ feedforward blocks, feedforward dimension $16 = 8 + 8$ and bit complexity $2\max\{\rho, \gamma\}B$. Moreover, each of $\mathcal{F}_l^1$ and $\mathcal{F}_l^2$ requires 1 feedforward block and bit complexity $\gamma$. The feedforward dimension of $\mathcal{F}_l^1$ is 5 where $\tilde{\mathcal{F}}_l^1$ incurs 4 from Lemma E.3 and 1 additional unit wipes out the (positive) carried value in 4-th coordinate from the skip-connection. The feedforward dimension of $\mathcal{F}_l^2$ is 3 where $\tilde{\mathcal{F}}_l^2$ incurs 1 and 2 additional units wipe out the carried values in 4-th and 7-th coordinates from the skip-connection. In total, each step $l \in [B]$ consists of $\max\{\rho, \gamma\} + 2$ feedforward blocks with skip-connections, feedforward dimension 16 and bit complexity $2\max\{\rho, \gamma\}B$.

The pre-processing network $\mathcal{F}_{pre}$ and the post-processing network $\mathcal{F}_{post}$ can be implemented with 1 feedforward block, feedforward dimension at most 2 (to carry $u$ and $w$ or $z_8$) and the bit complexity at most $\max\{\rho, \gamma\}B + 2$.

Thus, $\mathcal{N}_4$ consists of $(\max\{\rho, \gamma\} + 2)B + 2$ feedforward blocks with skip-connections, feedforward dimension 16 and bit complexity $2\max\{\rho, \gamma\}B$. $\qquad \square$

### A.5 POSITIONAL ENCODING

Consider the token id $\boldsymbol{u}^T \boldsymbol{X}^{(n)}$ in stage 1. If we define the positional encoding as

$$\boldsymbol{E} = r'\boldsymbol{u}[n-1, n-2, \cdots, 1, 0],$$

then we have

$$\boldsymbol{u}^T(\boldsymbol{X}^{(n)} + \boldsymbol{E}) = \boldsymbol{u}^T \boldsymbol{X}^{(n)} + r'[n-1, n-2, \cdots, 1, 0],$$

Since every element of $\boldsymbol{u}^T \boldsymbol{X}^{(n)}$ are bounded above by $r'$, positional encoding enforces the decreasing order used in stage 2 as the usual sequential order. Thus, the sequence id is no longer permutation equivariant. The upper bound on the magnitude of token ids increases by a factor of $n$, but it only affects parameter complexity and bit complexity logarithmically through $R$.

### A.6 PROOF OF THEOREM 3.1

The final Transformer network combines all 4 stages as $\mathcal{N} = \mathcal{N}_4 \circ \mathcal{N}_3 \circ \mathcal{N}_2 \circ \mathcal{N}_1$ where $\mathcal{N}_3$ and $\mathcal{N}_4$ applies the same function to each token independently. The mismatch in the embedding and feedforward dimensions is easily resolved by using the maximum dimension required and set all weights in the unused dimension to zero. We modify $\mathcal{N}_3$ to output both $u_{\lceil \frac{i}{B} \rceil}$ and $w_{\lceil \frac{i}{B} \rceil}$ as mentioned in the Remark A.4. We summarize all stages:

1. $\mathcal{N}_1$ projects the input sequence $\boldsymbol{X}^{(i)} \in \mathbb{R}^{d \times n}$ to the token ids $\boldsymbol{x}^{(i)} \in \mathbb{R}^n$. This stage consists of 1 feedforward block of dimension 1 and bit complexity $\lceil \log(2rn^2 N^2 d\sqrt{\pi} \delta^{-1}) \rceil \leq \log(r'\sqrt{d})$.

2. $\mathcal{N}_2$ further projects the token ids $\boldsymbol{x}^{(i)} \in \mathbb{R}^n$ to the sequence id $z^{(i)} \in \mathbb{R}$ permutation equivariantly. This stage consists of $2n$ Transformer blocks of attention dimension 1, feedforward dimension 4 and bit complexity $\lceil \log(2r'nN^2\sqrt{\pi}) \rceil \leq \log(R')$.

3. $\mathcal{N}_3$ combines a token id and a sequence id to obtain the contextual token id and finds two group strings. $N' \leq nN$ memorized contextual token ids are partitioned into $A$ groups of $B$ ids so that $N' \leq AB$. Two group strings are crafted as concatenations of contextual token ids and corresponding labels in the group. This stage consists of A feedforward blocks of dimension 8 and bit complexity $\lceil \max\{\log R, \log C\} \rceil B + \lceil \log R \rceil + 1$.

4. $\mathcal{N}_4$ extracts the correct label from the crafted strings. This stage consists of $(\max\{\rho, \gamma\} + 2)B + 2$ feedforward blocks of dimension 16 and bit complexity $2\lceil \max\{\log R, \log C\} \rceil B$.

In total, the network $\mathcal{N}$ uses $1 + 2n + A + (\max\{\rho, \gamma\} + 2)B + 2 = O(n + A + \max\{\rho, \gamma\}B)$ Transformer blocks of dimension 16 and bit complexity $\log(r'\sqrt{d}) + \log(R') + \lceil \max\{\log R, \log C\} \rceil B + \lceil \log R \rceil + 1 + 2\lceil \max\{\log R, \log C\} \rceil B = O(\log d + \lceil \max\{\log R, \log C\} \rceil B)$. We note that $R = 9r'R' \leq 150r'^2 nN^2 \leq 8000r^2 n^5 N^6 d\delta^{-2}$

Finally, we balance $A$ and $B$ as $A = \sqrt{nN \log(nN)}$ and $B = \sqrt{\frac{nN}{\log(nN)}}$ and conclude the proof of Theorem 3.1.

## B LARGE WIDTH AND FIXED BIT COMPLEXITY

In this section, we formally study the generalization of Theorem 3.1 in the case of large width (Section B.1) and fixed bit complexity (Section B.2) from Remark 3.3. To simplify the argument, we state and prove the result only for the case without permutation equivariance. The theorem for the permutation equivariance case is straightforwardly obtained from our stated theorem.

### B.1 LARGE WIDTH

We state and prove the theorem for large width.

**Theorem B.1.** *Assume the same setting as in Theorem 3.1. Let $L \leq \sqrt{nN}$. There exists a Transformer network $\mathcal{N} : \mathbb{R}^{d \times n} \to \mathbb{R}^{1 \times n}$ and positional encoding $\boldsymbol{E} \in \mathbb{R}^{d \times n}$ such that*

$$\mathcal{N}(\boldsymbol{X}^{(i)} + \boldsymbol{E}) = \boldsymbol{Y}^{(i)}$$

*for every $i \in [N]$. In both cases, the Transformer $\mathcal{N}$ has width $16nN/L^2$ ($m = 6nN/L^2$, $h = k = 1$ and $q = 16nN/L^2$), depth*

$$O\left(n + L\sqrt{\log(L)} + L\sqrt{\frac{1}{\log(L)}} \cdot \max\{\log C, \log R\}\right)$$

*and bit complexity bounded by*

$$O\left(\log d + L\sqrt{\frac{1}{\log(L)}} \cdot \max\{\log C, \log R\}\right)$$

*where we denote $R := 8000r^2\delta^{-2}dn^5N^6$.*

*Proof.* Stages 1 and 2 are the same as the proof of Theorem 3.1. For stages 3 and 4, instead of directly memorizing $nN$ contextual token ids directly, we construct $nN/L^2$ subnetworks where each memorize $L^2$ contextual token ids. By stacking subnetworks horizontally across width, we obtain the result. We remark that the width is not increased for the self-attention layers which are not used in the parallelized stages 3 and 4. $\qquad\square$

Theorem B.1 shows that, if depth is bounded above by $\tilde{O}(L)$ with $L > n$, then $\tilde{O}(d + n + nN/L)$ parameters are enough to memorize $N$ sequence classification examples of length $n$ with token dimension $d$. We count the number of parameters as linear in width instead of quadratic in width because our construction uses parallel $nN/L^2$ subnetworks without interaction among them.

## B.2 Fixed bit complexity

We state and prove the theorem for fixed bit complexity.

**Theorem B.2.** *Assume the same setting as in Theorem 3.1. Let $B \leq \sqrt{nN}$. There exists a Transformer network $\mathcal{N} : \mathbb{R}^{d \times n} \to \mathbb{R}^{1 \times n}$ and positional encoding $\boldsymbol{E} \in \mathbb{R}^{d \times n}$ such that*

$$\mathcal{N}(\boldsymbol{X}^{(i)} + \boldsymbol{E}) = \boldsymbol{Y}^{(i)}$$

*for every $i \in [N]$. In both cases, the Transformer $\mathcal{N}$ has width $16$ ($m = 6$, $h = k = 1$ and $q = 16$), depth*

$$O\left(n + \frac{nN}{B}\sqrt{\log(B)} + \frac{nN}{B}\sqrt{\frac{1}{\log(B)}} \cdot \max\{\log C, \log R\}\right)$$

*and bit complexity bounded by*

$$O\left(\log d + B\sqrt{\frac{1}{\log(B)}} \cdot \max\{\log C, \log R\}\right)$$

*where we denote $R := 8000r^2\delta^{-2}dn^5N^6$.*

*Proof.* Stage 1 and 2 are the same as the proof of Theorem 3.1. For stage 3 and 4, instead of directly memorizing $nN$ contextual token ids directly, we construct $nN/B^2$ subnetworks where each memorize $B^2$ contextual token ids. By stacking subnetworks vertically across depth, we obtain the result. $\qquad\square$

Theorem B.2 shows that, if bit complexity is bounded above by $\tilde{O}(B)$, then $\tilde{O}(d + n + nN/B)$ parameters are enough to memorize $N$ sequence classification examples of length $n$ with token dimension $d$. We count the number of parameters as linear in width instead of quadratic in width because our construction uses parallel $nN/L^2$ subnetworks without interaction among them.

# C Contextual Mapping Implications

## C.1 Sparse Attention Transformers

This section shows how our result generalizes to sparse-attention Transformers. Sparse-attention Transformers replaces self-attention subblocks with sparse counterparts. Let $\mathcal{A}_k^l \subset [n]$ be the $l$-th sparsity pattern of $k$-th token where $k \in [n], l \in [p]$. Given an input $\boldsymbol{Z} \in \mathbb{R}^{m \times n}$, the sparse self-attention subblock $F_l^{(SSA)}$ with $h$ heads and head size $k$ computes

$$F_l^{(SSA)}(\boldsymbol{Z}) = \boldsymbol{Z} + \sum_{i=1}^{h} \boldsymbol{W}_{l,i}^{(O)} \left( \boldsymbol{W}_{l,i}^{(V)} \boldsymbol{Z}_{\mathcal{A}_k^l} \right) \sigma_S \left[ \left( \boldsymbol{W}_{l,i}^{(K)} \boldsymbol{Z}_{\mathcal{A}_k^l} \right)^T \left( \boldsymbol{W}_{l,i}^{(Q)} \boldsymbol{Z}_{\mathcal{A}_k^l} \right) \right],$$

where $\boldsymbol{Z}_{\mathcal{A}_k^l} \in \mathbb{R}^{m \times |\mathcal{A}_k^l|}$ denote the submatrix consisting of columns of $Z$ in the index set $\mathcal{A}_k^l$. Again, $\boldsymbol{W}_{l,i}^{(O)} \in \mathbb{R}^{m \times k}$ and $\boldsymbol{W}_{l,i}^{(V)}, \boldsymbol{W}_{l,i}^{(K)}, \boldsymbol{W}_{l,i}^{(Q)} \in \mathbb{R}^{k \times m}$ are the weight matrices parametrizing the sparse self-attention subblock.

We make the following assumption on the sparsity pattern, which is the last condition among three conditions in Assumption 1 in Yun et al. (2020b). Thus, our assumption is strictly weaker.

**Assumption C.1.** (Relaxed version of Assumption 1 in Yun et al. (2020b)) Define a sequence of set $\{\mathcal{S}_k^t\}_{t \geq 1}$ as

$$\mathcal{S}_k^1 := \mathcal{A}_k^1, \mathcal{S}_k^t := \bigcup_{j \in \mathcal{A}_k^{(t-1) \mod p+1}} \mathcal{S}_j^{t-1}$$

We assume that the sparsity patterns $\{\mathcal{A}_k^l\}$ satisfy that there exists a finite $s \in \mathbb{N}$ such that

$$s = \min\{u | \mathcal{S}_k^u = [n] \text{ for all } k \in [n]\}.$$

We provide the sparse-attention version of our main result.

**Theorem C.1.** *Let* $N, d, n, C, s \in \mathbb{N}$ *and* $r \geq 1, 0 < \delta \leq 1$. *Let* $(\boldsymbol{X}^{(1)}, \boldsymbol{Y}^{(1)}), \cdots, (\boldsymbol{X}^{(N)}, \boldsymbol{Y}^{(N)}) \in \mathbb{R}^{d \times n} \times [C]^{1 \times n}$ *be* $N$ *input-output pairs of sequences that satisfies. Denote* $R := 8000 r^2 \delta^{-2} dn^5 N^6$. *Then, there exists a Transformer network* $F : \mathbb{R}^{d \times n} \to \mathbb{R}^{1 \times n}$ *with width 16 (*$m = 6$, $h = k = 1$ *and* $r = 16$*), depth*

$$O \left( ns + \sqrt{nN \log N} + \sqrt{\frac{nN}{\log N}} \cdot \max\{\log C, \log R\} \right)$$

*and bit complexity bounded by*

$$O \left( \log d + \sqrt{\frac{nN}{\log N}} \cdot \max\{\log C, \log R\} \right)$$

*such that* $F(\boldsymbol{X}^{(i)} \boldsymbol{P}) = \boldsymbol{Y}^{(i)} \boldsymbol{P}$ *for every* $i \in [N]$ *and for every permutation matrix* $\boldsymbol{P} \in \mathbb{R}^{n \times n}$.

*Proof.* The stage 1, 3 and 4 are the same as the proof of Theorem 3.1. In stage 2, each step of $\mathcal{N}_2$ in Lemma A.2 computes the maximum token id over the whole sequence using 1 self-attention layer. Under Assumption C.1, we instead can compute the maximum token id over the allowed sparsity pattern. Since the whole sequence is covered within a recursion of $s$ consecutive sparsity patterns and taking maximum is associative, repeating $s$ self-attention layer will give the desired maximum token id over the whole sequence again. Now, the other component in stage 2 works as before and the resulting memorization is achieved in the same way. The only overhead in this approach is the $s$ times larger number of self-attention layers. □

The simplicity of our contextual mapping enables easy generalization to the sparse attention. Since the sparse attention Transformer only constrains the sparsity pattern in the self-attention subblock, stages 1, 3 and 4 in our construction works without any modification. For the contextual mapping in stage 2, we achieve the same memorization capacity with only $s$ times more self-attention layers. The number of parameters is $\tilde{O}(d + sn + \sqrt{N})$.

C.2 IMPROVED CONTEXTUAL MAPPING FOR FUNCTION APPROXIMATION

Our idea can be used to reduce the number of self-attention layers for the contextual mapping in function approximation settings, too. We recall the formal definition of the contextual mapping as follows.

**Definition C.2.** (Definition 3.1 in Yun et al. (2020a), Contextual Mapping) Consider a finite set $\mathcal{L} \subset \mathbb{R}^{d \times n}$. A map $q : \mathcal{L} \to \mathbb{R}^{1 \times n}$ is a contextual mapping if the map satisfies the following:

1. For any $L \in \mathcal{L}$, the $n$ entries in $q(L)$ are distinct.

2. For any $L, L' \in \mathcal{L}$ with $L \neq L'$, all entries of $q(L)$ and $q(L')$ are distinct.

We state our improvement of the contextual mapping.

**Theorem C.2.** *(Improved version of Lemma 6 in Yun et al. (2020a)) Consider the following subset of $\mathcal{G}_\delta = \{0, \delta, \cdots, 1 - \delta\}^{d \times n}$:*

$$\tilde{\mathcal{G}}_\delta := \{L \in \mathcal{G}_\delta | L_{:,i} \neq L_{:,j} \text{ for all } i \neq j\}.$$

*Assume that $n \geq 2$ and $\delta^{-1} \geq 2$. Then, there exist a function $g_c : \mathbb{R}^{4 \times n} \to \mathbb{R}^{4 \times n}$ composed of $3n$ Transformer blocks with $h = 1$, $k = 1$ and $q = 4$ that employ the hardmax operator, vectors $\boldsymbol{w} \in \mathbb{R}^d, \boldsymbol{u} \in \mathbb{R}^4$, constants $t_l, t_r \in \mathbb{R}$ ($0 < t_l < t_r$), such that $q(L) := \boldsymbol{u}^T g_c(\boldsymbol{w}^T L, \boldsymbol{w}^T L, \boldsymbol{0}, \boldsymbol{0})$ satisfies the following propeties:*

1. *For any $L \in \tilde{\mathcal{G}}_\delta$, the entries of $q(L)$ are all distinct.*

2. *For any $L, L' \in \tilde{\mathcal{G}}_\delta$ such that $L$ is not a permutation of $L'$, all entries of $q(L), q(L')$ are distinct.*

3. *For any $L \in \tilde{\mathcal{G}}_\delta$, all the entries of $q(L)$ are in $[t_l, t_r]$.*

4. *For any $L \in \mathcal{G}_\delta^+ \setminus \tilde{\mathcal{G}}_\delta$, all the entries of $q(L)$ are outside $[t_l, t_r]$.*

*Proof.* Since token embeddings are $\delta$-discretized, we can concatenate each coordinate to obtain the token id as in Yun et al. (2020a). Let $\boldsymbol{w}$ be the vector that represents such concatenation as a linear operation. As in stage 2 of our proof for Theorem 3.1, we concatenate the token id in the decreasing order of magnitude to obtain the sequence id in $n$ steps. Then, we set $\boldsymbol{u}$ appropriately to obtain the "contextual token id" or the concatenation of token id and sequence id in $q(L)$. Then, the first three conditions are easy to check. The first condition is trivially true because distinct token will have distinct token id and consequently distinct contextual token id. The second condition is also true because if $L$ is not a permutation of $L'$, then their sequence ids should differ. The third condition is true since a linear function in a compact region is bounded.

Consider the final condition. In any step of our efficient contextual mapping, when the maximum token id is zero, there must be duplicate tokens in the input sequence. Conversely, if there are duplicate tokens in the input sequence, the maximum token id should be zero at some steps of our efficient contextual mapping. We may use one more feedforward block in each step of the efficient contextual mapping to subtract the sequence id by $M$ if such zero maximum token id is observed. Here, $M$ is the maximum value possible for the sequence id. Then, the sequence id will still be negative in the end so $q(L)$ will also be negative. Thus, the final condition is also true. □

We highlight the difference in the architecture. The major difference is the number of Transformer blocks used. We use $3n$ layers (linear in the sequence length) while Yun et al. (2020a) use $\delta^{-d} + 1$ layers (exponential in the embedding dimension). Since the sequence length and the embedding dimension are of the same order in practice, our construction exponentially improves Lemma 6 in Yun et al. (2020a). The minor difference in the architecture is the intermediate embedding dimension (4 in ours and $d$ in Yun et al. (2020a)) and the number of attention heads (1 in ours and 2 in Yun et al. (2020a)).

## D  OTHER TASKS

We provide the proof for Theorem 4.1 (Section D.1) and formal results on language modeling tasks. We consider two language modeling tasks that are commonly used to pre-training Transformers: masked language modeling (Section D.2) and autoregressive language modeling (Section D.3).

### D.1  PROOF OF THEOREM 4.1

Stages 1 and 2 are the same as the proof of Theorem 3.1. Instead of classifying the contextual token id, we can directly classify sequence ids in stages 3 and 4. Since there are $N$ possible sequence id, we replace $nN$ with $N$ in the parameter complexity from Theorem 3.1.

### D.2  MASKED LANGUAGE MODELING

Let $\boldsymbol{X} \in \mathbb{R}^{d \times T}$ and $\boldsymbol{Y} \in [V]^T$ be corpus data of $T$ tokens represented as embedding vectors and token ids, respectively. The corpus data is divided into $P = T - n + 1$ sequences $\boldsymbol{X}^{(1)}, \cdots, \boldsymbol{X}^{(P)} \in \mathbb{R}^{d \times n}$ and $\boldsymbol{Y}^{(1)}, \cdots, \boldsymbol{Y}^{(P)} \in [V]^n$ of length $n$ by taking $\boldsymbol{X}^{(i)} = \boldsymbol{X}[:, i : i+n]$ and $\boldsymbol{Y}^{(i)} = \boldsymbol{Y}[i : i+n]$. Here, we use the slice index notation $i : j$ for items from $i$ (inclusive) to $j$ (exclusive). We mask $m$ out of $n$ tokens in the sequence with $Q = \binom{n}{m}$ possible masking patterns $\boldsymbol{M}^{(1)}, \cdots, \boldsymbol{M}^{(Q)} \in \{0, 1\}^n$. We define the *masked sequences $\boldsymbol{M}^{(j)} \circ \boldsymbol{X}^{(i)}$ extracted from $\boldsymbol{X}$* as the sequence $\boldsymbol{X}^{(i)}$ with columns masked according to the pattern $\boldsymbol{M}^{(j)}$.

We say that a Transformer network $\mathcal{N} : \mathbb{R}^{d \times n} \to \mathbb{R}^{1 \times n}$ and positional encoding $\boldsymbol{E} \in \mathbb{R}^{d \times n}$ *memorizes masked language modeling* of $\boldsymbol{X}, \boldsymbol{Y}$ if

$$\mathcal{N}(\boldsymbol{M}^{(j)} \circ \boldsymbol{X}^{(i)} + \boldsymbol{E}) = \boldsymbol{Y}^{(i)}$$

for every $i \in [P], j \in [Q]$.

**Theorem D.1.** *Let $T, d, n, m, V \in \mathbb{N}$ and $r \geq 1, 0 < \delta \leq 1$. Let $\boldsymbol{X} \in \mathbb{R}^{d \times T}, \boldsymbol{Y} \in [V]^T$ be a corpus data of $T$ tokens represented as embedding vectors and token ids, respectively. Suppose that the masked sequences extracted from $\boldsymbol{X}$ are distinct and tokenwise $(r, \delta)$-separated.*

*Then, there exists a Transformer network $\mathcal{N} : \mathbb{R}^{d \times n} \to \mathbb{R}^{1 \times n}$ and positional encoding $\boldsymbol{E} \in \mathbb{R}^{d \times n}$ that memorizes masked language modeling of $\boldsymbol{X}, \boldsymbol{Y}$. The Transformer $\mathcal{N}$ has width 16 ($m = 6$, $h = k = 1$ and $q = 16$), depth*

$$O\left(n + \sqrt{nPQ \log(nPQ)} + \sqrt{\frac{nPQ}{\log(nPQ)} \cdot \max\{\log V, \log R\}}\right)$$

*and bit complexity bounded by*

$$O\left(\log d + \sqrt{\frac{nPQ}{\log(nPQ)} \cdot \max\{\log V, \log R\}}\right)$$

*where we denote $P = T - n + 1$, $Q = \binom{n}{m}$ and $R := 8000 r^2 \delta^{-2} d n^5 P^6 Q^6$.*

*Proof.* We apply Theorem 3.1 to memorize $PQ$ masked sequences $\boldsymbol{M}^{(j)} \circ \boldsymbol{X}^{(i)}$.  □

### D.3  AUTOREGRESSIVE LANGUAGE MODELING

Let $\boldsymbol{X} \in \mathbb{R}^{d \times T}$ and $\boldsymbol{Y} \in [V]^T$ be a corpus data of $T$ tokens represented as embedding vectors and token ids, respectively. The corpus data is divided into $P = T - n$ sequences $\boldsymbol{X}^{(1)}, \cdots, \boldsymbol{X}^{(P)} \in \mathbb{R}^{d \times n}$ and $y^{(1)}, \cdots, y^{(P)} \in [V]$ of length $n$ by taking $\boldsymbol{X}^{(i)} = \boldsymbol{X}[:, i : i + n]$ and $y^{(i)} = \boldsymbol{Y}[i + n]$. We call $\boldsymbol{X}^{(i)}$ as the *input sequences extracted from $\boldsymbol{X}$*.

We say that a Transformer network $\mathcal{N} : \mathbb{R}^{d \times n} \to \mathbb{R}^{1 \times n}$ and positional encoding $\boldsymbol{E} \in \mathbb{R}^{d \times n}$ *memorizes autoregressive language modeling* of $\boldsymbol{X}, \boldsymbol{Y}$ if

$$\mathcal{N}(\boldsymbol{X}^{(i)} + \boldsymbol{E}) = y^{(i)}$$

for every $i \in [P]$.

**Theorem D.2.** *Let $T, d, n, m, V \in \mathbb{N}$ and $r \geq 1, 0 < \delta \leq 1$. Let $\boldsymbol{X} \in \mathbb{R}^{d \times T}, \boldsymbol{Y} \in [V]^T$ be a corpus data of $T$ tokens represented as embedding vectors and token ids, respectively. Suppose that the input sequences extracted from $\boldsymbol{X}$ are distinct and tokenwise $(r, \delta)$-separated.*

*Then, there exists a Transformer network $\mathcal{N} : \mathbb{R}^{d \times n} \to \mathbb{R}^{1 \times n}$ and positional encoding $\boldsymbol{E} \in \mathbb{R}^{d \times n}$ that memorizes autoregressive language modeling of $\boldsymbol{X}, \boldsymbol{Y}$. The Transformer $\mathcal{N}$ has width 16 ($m = 6, h = k = 1$ and $q = 16$), depth*

$$O\left(n + \sqrt{P \log P} + \sqrt{\frac{P}{\log P}} \cdot \max\{\log V, \log R\}\right)$$

*and bit complexity bounded by*

$$O\left(\log d + \sqrt{\frac{P}{\log P}} \cdot \max\{\log V, \log R\}\right)$$

*where we denote $P = T - n$ and $R := 8000 r^2 \delta^{-2} d n^5 P^6$.*

*Proof.* We apply Theorem 4.1 to memorize $P$ input sequences $\boldsymbol{X}^{(i)}$. $\square$

## E    TECHNICAL LEMMAS

Here, we state technical lemmas that are used in our proofs.

**Lemma E.1.** *(Lemma 13 from Park et al. (2021)) Let $N, d \in \mathbb{N}$ and $\boldsymbol{x}^{(1)}, \cdots \boldsymbol{x}^{(N)} \in \mathbb{R}^d$. Then, there exists a unit vector $\boldsymbol{u} \in \mathbb{R}^d$ such that $\frac{1}{N^2}\sqrt{\frac{8}{\pi d}}\|\boldsymbol{x}^{(i)} - \boldsymbol{x}^{(j)}\| \leq |\boldsymbol{u}^T(\boldsymbol{x}^{(i)} - \boldsymbol{x}^{(j)})| \leq \|\boldsymbol{x}^{(i)} - \boldsymbol{x}^{(j)}\|$ for every $i, j \in [N]$.*

**Lemma E.2.** *Let $n \in \mathbb{N}$ and $r', P > 1$. Then, there exists a neural network $\mathcal{F} : \mathbb{R}^{1 \times n} \to \mathbb{R}^{1 \times n}$ consisting of a single softmax self-attention with 1 head, head size 1 and bit complexity $\lceil\log\left(\log(8n^{3/2}r'P)\right)\rceil$ such that $\mathcal{F}(\boldsymbol{x}^T) = c\boldsymbol{1}_n^T$ with $x_{\max} - \frac{1}{2P\sqrt{n}} \leq c \leq x_{\max}$ whenever $\boldsymbol{x} \in \mathbb{R}^n$ satisfies*

- *$|\boldsymbol{x}[i]| \leq 2r'$ for $i \in [n]$ and*
- *$\boldsymbol{x}[i] \leq x_{\max} - 2$ for $i \in [n]$ with $\boldsymbol{x}[i] \neq x_{\max}$,*

*where we denote $x_{\max} = \left(\max_{i \in [n]} \boldsymbol{x}[i]\right)$.*

*Proof.* When we have hardmax activation on the attention matrix, it is easy to construct the network that satisfies the condition. Consider the following self-attention:

$$\mathcal{F}(\boldsymbol{x}) = 1 \cdot (1 \cdot \boldsymbol{x})\sigma_H\left[(1 \cdot \boldsymbol{x})^T(0 \cdot \boldsymbol{x} + \boldsymbol{1}_n^T)\right] = \boldsymbol{x}\sigma_H\left[\boldsymbol{x}^T\boldsymbol{1}_n^T\right] = \left(\max_{i \in [n]} x_i\right)\boldsymbol{1}_n^T.$$

Indeed, the output is exactly $x_{\max}$ and the bit complexity is 1 since all weights are either 0 or 1.

To approximate hardmax with softmax, we introduce some large factor $t > 0$ in the attention matrix. Consider the following self-attention:

$$\mathcal{F}(\boldsymbol{x}) = 1 \cdot (1 \cdot \boldsymbol{x})\sigma_S\left[(t \cdot \boldsymbol{x})^T(0 \cdot \boldsymbol{x} + \boldsymbol{1}_n^T)\right] = \boldsymbol{x}\sigma_S\left[t\boldsymbol{x}^T\boldsymbol{1}_n^T\right] = c\boldsymbol{1}_n^T$$

where

$$c = \sum_{i=1}^n \boldsymbol{x}[i] \frac{\exp(t\boldsymbol{x}[i])}{\sum_{j=1}^n \exp(t\boldsymbol{x}[j])}.$$

Since $c$ is a convex combination of $\boldsymbol{x}[i]$'s, it is easy to see that $x_{\max}$ upper bounds $c$. It suffices to find $t$ that satisfies the lower bound condition.

Choose $t = \lceil \frac{1}{2} \log(8n^{3/2} r' P) \rceil$. We lower bound the softmax weights on $x_{\max}$ as

$$
\begin{aligned}
p_{\max} &:= \frac{\sum_{i: \boldsymbol{x}[i] = x_{\max}} \exp(t\boldsymbol{x}[i])}{\sum_{i=1}^{n} \exp(t\boldsymbol{x}[i])} \\
&= \frac{\sum_{i: \boldsymbol{x}[i] = x_{\max}} \exp(t\boldsymbol{x}[i])}{\sum_{i: \boldsymbol{x}[i] = x_{\max}} \exp(t\boldsymbol{x}[i]) + \sum_{i: \boldsymbol{x}[i] \neq x_{\max}} \exp(t\boldsymbol{x}[i])} \\
&\geq \frac{\sum_{i: \boldsymbol{x}[i] = x_{\max}} \exp(t x_{\max})}{\sum_{i: \boldsymbol{x}[i] = x_{\max}} \exp(t x_{\max}) + \sum_{i: \boldsymbol{x}[i] \neq x_{\max}} \exp(t(x_{\max} - 2))} \\
&= \frac{n_{\max}}{n_{\max} + (n - n_{\max}) \exp(-2t)} \\
&= \frac{1}{1 + (\frac{n}{n_{\max}} - 1) \exp(-2t)} \\
&\geq \frac{1}{1 + (\frac{n}{n_{\max}} - 1) \frac{1}{8n^{3/2} r' P}} \\
&\geq \frac{1}{1 + \frac{1}{8 r' P \sqrt{n}}},
\end{aligned}
$$

where $n_{\max} := |\{i : \boldsymbol{x}[i] = x_{\max}\}|$. Now, we can lower bound $c$ as

$$
\begin{aligned}
c &\geq x_{\max} p_{\max} - 2r'(1 - p_{\max}) \\
&= x_{\max} - (x_{\max} + 2r')(1 - p_{\max}) \\
&\geq x_{\max} - 4r'(1 - p_{\max}) \\
&\geq x_{\max} - 4r'\left(1 - \frac{1}{1 + \frac{1}{8 r' P \sqrt{n}}}\right) \\
&= x_{\max} - \frac{\frac{1}{2 P \sqrt{n}}}{1 + \frac{1}{8 r' P \sqrt{n}}} \\
&\geq x_{\max} - \frac{1}{2 P \sqrt{n}}.
\end{aligned}
$$

This self-attention module only has 1 head with head size 1. All weights are either $0$, $1$ or $t$ so the bit complexity is $\lceil \log t \rceil \leq \lceil \log (\log(8n^{3/2} r' P)) \rceil$. $\qquad\square$

**Lemma E.3.** *Let $r' \in \mathbb{N}$. Then, there exists a neural network $\mathcal{F} : \mathbb{R}^2 \to \mathbb{R}$ with 1 hidden layer, width 4 and bit complexity $\lceil \log 2r' \rceil$ such that*

$$
\mathcal{F}(x, y) = \begin{cases} r' & \text{if } |x - y| < \frac{1}{2} \\ 0 & \text{if } |x - y| > 1 \end{cases}.
$$

*Proof.* Consider the following neural network:

$$
\mathcal{F}(x, y) = [r' \quad -r' \quad -r' \quad r'] \sigma_R \left( \begin{bmatrix} 2 & -2 \\ 2 & -2 \\ 2 & -2 \\ 2 & -2 \end{bmatrix} \begin{bmatrix} x \\ y \end{bmatrix} + \begin{bmatrix} 2 \\ 1 \\ -1 \\ -2 \end{bmatrix} \right)
$$

$$
= r' \left( \sigma_R(2x - 2y + 2) - \sigma_R(2x - 2y + 1) - \sigma_R(2x - 2y - 1) + \sigma_R(2x - 2y - 2) \right).
$$

It is straightforward to see that the network $\mathcal{F}$ computes the desired function. This network has 1 hidden layer and width 4. All parameters are either $\pm 1$, $\pm 2$ or $\pm r'$ so the bit complexity is $\lceil \log 2r' \rceil$. $\qquad\square$

**Lemma E.4.** *Let $a, b, w \in \mathbb{N}$ with $a < b$. Then, there exists a neural network $\mathcal{F} : \mathbb{R} \to \mathbb{R}$ with 1 hidden layer, width 4 and bit complexity $\lceil \log w \rceil + \lceil \log(2b + 1) \rceil$ such that*

$$
\mathcal{F}(x) = \begin{cases} w & \text{if } x \in [a, b] \\ 0 & \text{if } x \notin [a - \frac{1}{2}, b + \frac{1}{2}] \end{cases}.
$$

*Proof.* Consider the following neural network:

$$
\mathcal{F}(x, y) = \begin{bmatrix} w & -w & -w & w \end{bmatrix} \sigma_R \left( \begin{bmatrix} 2 \\ 2 \\ 2 \\ 2 \end{bmatrix} x - \begin{bmatrix} 2a - 1 \\ 2a \\ 2b \\ 2b + 1 \end{bmatrix} \right)
$$

$$
= w \left( \sigma_R(2x - 2a + 1) - \sigma_R(2x - 2a) - \sigma_R(2x - 2b) + \sigma_R(2x - 2b - 1) \right).
$$

It is straightforward to see that the network $\mathcal{F}$ computes the desired function. This network has 1 hidden layer and width 4. All parameters are either $\pm w, 2$ or $2a - 1, 2a, 2b, 2b + 1$ so the bit complexity is $\lceil \log w \rceil + \lceil \log(2b + 1) \rceil$. $\qquad\square$

**Lemma E.5.** *Let $i, j, n, x, y \in \mathbb{N}$ with $i < j \leq n$. Denote $\phi(z) = \sigma_R(\sigma_R(2z) - \sigma_R(4z - 2))$. Then, there exists a neural network $\mathcal{F} : \mathbb{R}^3 \to \mathbb{R}^3$ consisting of $j - i + 1$ feedforward blocks with skip-connections every 2 layers, feedforward dimension 8 and bit complexity $2n$ such that*

$$
\mathcal{F}\left( \begin{bmatrix} \phi^{(i-1)}\left( \frac{x}{2^n} + \frac{1}{2^{n+1}} \right) \\ \phi^{(i-1)}\left( \frac{x}{2^n} + \frac{1}{2^{n+2}} \right) \\ y \end{bmatrix} \right) = \begin{bmatrix} \phi^{(j)}\left( \frac{x}{2^n} + \frac{1}{2^{n+1}} \right) \\ \phi^{(j)}\left( \frac{x}{2^n} + \frac{1}{2^{n+2}} \right) \\ y + \mathrm{BIN}_{i:j}(x) \end{bmatrix}.
$$

*Proof.* We construct $\mathcal{F} = \mathcal{F}_j \circ \mathcal{F}_{j-1} \circ \cdots \circ \mathcal{F}_i$ as a composition of $j - i + 1$ feedforward blocks $\mathcal{F}_l : \mathbb{R}^2 \to \mathbb{R}^2, i \leq l \leq j$ where each feedforward block contains a single feedforward layer with 8 hidden units and a skip-connection. In step $l$, the block $\mathcal{F}_l$ computes

$$
\begin{bmatrix} \phi^{(l-1)}\left( \frac{x}{2^n} + \frac{1}{2^{n+1}} \right) \\ \phi^{(l-1)}\left( \frac{x}{2^n} + \frac{1}{2^{n+2}} \right) \\ y \end{bmatrix} \mapsto \begin{bmatrix} \phi^{(l)}\left( \frac{x}{2^n} + \frac{1}{2^{n+1}} \right) \\ \phi^{(l)}\left( \frac{x}{2^n} + \frac{1}{2^{n+2}} \right) \\ y + 2^{j-l}\mathrm{BIN}_l(x) \end{bmatrix}. \tag{2}
$$

Since we have

$$
\mathrm{BIN}_{i:j}(x) = \sum_{l=i}^{j} 2^{j-l}\mathrm{BIN}_l(x),
$$

the block computation in Equation 2 ensures that $\mathcal{F}$ computes the desired function.

**Construction of $\mathcal{F}_l$.** To obtain Equation 2, we define the feedforward block $\mathcal{F}_l$ with a skip-connection as

$$
\mathcal{F}_l\left( \begin{bmatrix} z_1 \\ z_2 \\ z_3 \end{bmatrix} \right) = \begin{bmatrix} z_1 \\ z_2 \\ z_3 \end{bmatrix} + \begin{bmatrix} \phi(z_1) - z_1 \\ \phi(z_2) - z_2 \\ f_l(z_1, z_2) \end{bmatrix} = \begin{bmatrix} \phi(z_1) \\ \phi(z_2) \\ z_3 + f_l(z_1, z_2) \end{bmatrix}
$$

where

$$
f_l(z_1, z_2) = 2^{j-l}\left( 2^{n+1-l}\phi(z_2) - 2^{n+1-l}\phi(z_1) + \frac{1}{2} \right).
$$

We first note that the triangle function $\phi$ can be implemented with one hidden layer as

$$
\phi(z) = \sigma_R(2z) - \sigma_R(4z - 2) + \sigma_R(2z - 2).
$$

Also, we can implement identity function as

$$
z = \frac{1}{2}\sigma_R(2z) - \frac{1}{2}\sigma_R(-2z).
$$

Thus, the following 8 hidden units

$$
\sigma_R(2z_1), \sigma_R(4z_1 - 2), \sigma_R(2z_1 - 2), \sigma_R(-2z_1)
$$
$$
\sigma_R(2z_2), \sigma_R(4z_2 - 2), \sigma_R(2z_2 - 2), \sigma_R(-2z_2)
$$

are enough to represent $\mathcal{F}_l$.

We check Equation 2 as follows:

$$
\mathcal{F}_l \left( \begin{bmatrix} \phi^{(l-1)}\left(\frac{x}{2^n}+\frac{1}{2^{n+1}}\right) \\ \phi^{(l-1)}\left(\frac{x}{2^n}+\frac{1}{2^{n+2}}\right) \\ y \end{bmatrix} \right) = \begin{bmatrix} \phi\left(\phi^{(l-1)}\left(\frac{x}{2^n}+\frac{1}{2^{n+1}}\right)\right) \\ \phi\left(\phi^{(l-1)}\left(\frac{x}{2^n}+\frac{1}{2^{n+2}}\right)\right) \\ y + f_l\left(\phi^{(l-1)}\left(\frac{x}{2^n}+\frac{1}{2^{n+1}}\right), \phi^{(l-1)}\left(\frac{x}{2^n}+\frac{1}{2^{n+2}}\right)\right) \end{bmatrix}
$$

$$
= \begin{bmatrix} \phi^{(l)}\left(\frac{x}{2^n}+\frac{1}{2^{n+1}}\right) \\ \phi^{(l)}\left(\frac{x}{2^n}+\frac{1}{2^{n+2}}\right) \\ y + 2^{j-l}\text{BIN}_l(x) \end{bmatrix}
$$

because

$$
\text{BIN}_l(x) = 2^{n+1-l}\phi^{(l)}\left(\frac{x}{2^n}+\frac{1}{2^{n+2}}\right) - 2^{n+1-l}\phi^{(l)}\left(\frac{x}{2^n}+\frac{1}{2^{n+1}}\right) + \frac{1}{2}.
$$

Finally, since all parameters are have the form $2^k$ for some $-1 \leq k \leq 2n$, the bit complexity is $2n$. $\qquad\square$

## F  EXPERIMENTAL SETUP

We use HuggingFace [9] PyTorch implementation of the BERT model for our experiments. All experiments are conducted on an Nvidia Quatro RTX 5000, 16 GB memory GPU in a machine with Intel(R) Xeon(R) Silver 4214 CPU @ 2.20GHz.

As mentioned in the main paper, we use 14,000 random samples in the named entity recognition dataset from CoNLL-2003 (Tjong Kim Sang & De Meulder, 2003) for token classification and 50,000 random samples in the MNLI dataset from GLUE benchmark (Wang et al., 2019) for sequence classification. For token classification, the task is to classify the named entity type for each token among 9 possible classes. The sequence classification dataset aims to classify the relationship between sentence pairs as 3 classes: *entailment, contradiction*, and *neutral*. We vary the dataset size by randomly order examples and picking first $p\%$ for $p = 10, 20, \cdots, 100$.

We vary the model size through the embedding size $m$, which is varied by 12 and 96 for token and sequence classification tasks, respectively. As mentioned in the main paper, we fix the number of layers as $L = 6$, the number of attention head as $h = 12$, the embedding to head size ratio as $m/k = h = 12$ and the feedforward to embedding size ratio as $q/m = 4$, as commonly done in practice.

We optimize using Adam optimizer (Kingma & Ba, 2015) with learning rate 0.00002, batch size 32 and dropout rate 10%. We train our models for 1,500 and 7,500 steps for token and sequence classification, respectively. We determine the above number of steps to ensure that the training error does not improve at least for the last 3 epochs.

For Figure 2, we choose the minimum size memorizing model as the smallest model that reaches the training error 0.005. The maximum training errors of selected models are 0.00499 and 0.00450 for token and sequence classification tasks, respectively. The average training errors of selected models are 0.00464 and 0.00259 for token and sequence classification tasks, respectively.

---

[9] https://huggingface.co/

