# OpenReview forum: "Provable Memorization Capacity of Transformers"
_ICLR.cc/2023/Conference — ICLR 2023 poster_

### Official Review · Reviewer_YzgV · 2022-10-23

**Confidence:** 3
**Correctness:** 4
**Technical Novelty And Significance:** 3
**Empirical Novelty And Significance:** Not applicable
**Recommendation:** 8

**Clarity, Quality, Novelty And Reproducibility:**

The paper is cleanly written and pleasant to read.
Proofs and experiment details are provided in the appendix.

A minor clarification question: $L$ is defined das the depth of the Transformer, but the depth is also $\tilde{O}(n+L)$ in Remark 3.3. Does this mean that given a depth-$L$ Transformer, we can modify it into another wider Transformer with depth $\tilde{O}(n+L)$?

**Strength And Weaknesses:**

In addition to the upper bound in main result, the paper provides remarks on:
- extending the results to real-valued / vector-valued output;
- allowing wider width (the main result has width=16) or bounded bit complexity (the main result has bit complexity growing roughly as $O(\sqrt{nN})$);
- connection to fully-connected ReLU net, i.e. the bound if we apply Vardi et al. 2022 directly.

I have no major complaint about the technical part of the paper. The main novelty over Vardi et al. 2022 comes from the use of self-attention blocks, which allows parameter sharing across tokens in a sequence, saving a factor linear in the sequence length.
I should note though that even though I read Vardi et al., I'm not well versed with the literature so may miss some subtleties.

Some minor comments about the experiments:
- Fig 1: how many runs are performed for each cell of the heatmap? Could you increase the granularity of Fig 1(b) to show a clearer trend?
- Fig 2: do the points correspond to models that have reached 0 (or near 0) training error? How many runs are there for each point? If there's only 1 run, then the trends may not be trustworthy. Please consider using multiple runs and show the median and standard error.


**Summary Of The Paper:**

This paper gives an upper bound on the number of parameters required to remember $N$ length-$n$ sequences, where each token is of dimension $d$.
The parameter upper bound is $\tilde{O}(d + n + \sqrt{nN})$. Together with bit complexity $\tilde{O}(\sqrt{nN})$, the upper bound is optimal up to log factors in terms of bit counts.

The proof technique is closely related to that of Vardi et al. 2022, with novelty lying in the use of self-attention, which improves parameter efficiency. As a comparison, directly applying the result in Vardi et al. (which is for fully-connected ReLU nets) will give a parameter count of $\tilde{O}(dn + \sqrt{nN})$, i.e. there is a factor of $dn$ as opposed to $d+n$ as in this paper.

The paper also provides a similar upper bound for sequence classification setup (i.e. the label is a scalar, rather than a sequence as in the main result), as well as empirical evidence that the parameter counts roughly follow the trend predicted by theory.

**Summary Of The Review:**

To my knowledge, this paper is the first to provide memorization capacity of Transformers with finite precision. The paper is well written and the results are of interests to the community, so I recommend an accept.

---

> ### Author Response · Authors · 2022-11-18
> **Response to Reviewer YgzV**
>
> We sincerely appreciate the time and effort that each reviewer spent on thoughtful reviews and comments. We address your comments below:
>
> > Fig 1: how many runs are performed for each cell of the heatmap? Could you increase the granularity of Fig 1(b) to show a clearer trend?
>
> Each cell of the heatmap has a value from a single run. We agree that the increased granularity and runs would better represent trend. Unfortunately, we did not have enough time and resources to perform more experiments and increase the granularity of Figure 1(b).
>
> > Fig 2: do the points correspond to models that have reached 0 (or near 0) training error? How many runs are there for each point? If there's only 1 run, then the trends may not be trustworthy. Please consider using multiple runs and show the median and standard error.
>
> Yes, we chose the smallest models that reached near 0 training error (0.005 or 0.5%). Again, each point is a value from a single run. We agree that the trends is not completely representative. Unfortunately, we did not have enough time and resources to finish our experiments to increase the number of runs in Figure 2.
>
> From the smooth change of training errors in our heatmap and the granularity of our search space, we claim that the current linear trend at least provide complementary observation to our theory. We leave more detailed empirical test as a future work.
>
> > A minor clarification question: $L$ is defined das the depth of the Transformer, but the depth is also $\tilde{O}(n+L)$ in Remark 3.3. Does this mean that given a depth-$L$ Transformer, we can modify it into another wider Transformer with depth $\tilde{O}(n+L)$?
>
> In remark 3.3, $L$ is used as a parameter to control the depth, not the depth itself. Since we need at least $n$ self-attention layers for contextual mapping, when $L > n$, you may interpret the depth as $\tilde{O}(L)$. Please see Theorem B.1 in page 24 and the directly following discussion.

---

### Official Review · Reviewer_Nzx7 · 2022-10-25

**Confidence:** 3
**Correctness:** 3
**Technical Novelty And Significance:** 3
**Empirical Novelty And Significance:** 2
**Recommendation:** 5

**Clarity, Quality, Novelty And Reproducibility:**

The theoretical statements have been clearly presented. However, as suggested above, the novelty of the work is difficult to check because of a lack of a proper proof sketch in the main paper or the appendix.

**Strength And Weaknesses:**

The major strength of the paper is the parameter efficient transformer (which is nearly optimal in the number of bit counts) constructed for memorizing $N$ sequences. Furthermore, the authors show experiments verifying some of their claims.

However, I feel that the use of the self-attention module in the construction hasn't been highlighted in the paper and is difficult to read from the proof in the appendix (please see more questions below). One of the major questions that I have is whether we can use the result of [1] in two steps: first, we project each token in the $n$ length sequence into a single bit, and then further project the $n$ dimensional representation into a single bit.

A section similar to 3.1 in [1] would have helped a lot.

Moreover, I have the following questions:

(a) Can the authors explain the role of the self-attention module inside the construction? Can one simply use a feed-forward network in place of the self-attention module? My reasoning is that after stage 1, you only need to store a single bit for each token in the sequence. Hence, is it possible to use a feedforward layer of dimension $n$ that "approximates maximum token id among the
first coordinate values and outputs in the second coordinate"?

If there is a difference, this can be used to compare attention models with MLP-only models introduced in [2, 3].

(b) Can the authors provide a discussion on how the contextual mapping used in Stage 2 improves over the selective shifting-based contextual mapping from Yun et al. (2020a)? Does the pre-processing in Stage 1 help reduce the overhead in Stage 2 for contextual mapping? Also, can the authors provide some discussion on how their contextual mapping is capable of "incorporating sparse self-attention settings with minimal parameter overhead"?

(c) In figure 2, how do the authors observe a square root relation between training data size and model size? The plots seem very linear. In plot 2(b), if we remove the point (30000, 2.5e7) which could have been a noisy point, the plot will become more linear.

(d) The experiments have been conducted with a fixed hyperparameter set (batch size, learning rate, etc.). That implies the plots in 2(a) and (b) could be highly dependent on the hyperparameters used. How do the plots look with a different set of hyperparameters?

(e) Is there a relation between the entropy observed in the experiments in Figure 3 and the number of sequences $N$? (By theorem 3.1, there should be a square root dependence between bit complexity and $N$).

(f) How do the plots in Figures 2 and 3 change with changes in the length of the sequences?

(g) The construction doesn't require the number of attention heads to change with a change in the number of sequences. Do the authors observe the same in their experiments, i.e. the behavior of the training loss doesn't depend on the number of attention heads used?


1: On the Optimal Memorization Power of ReLU Neural Networks. Gal Vardi, Gilad Yehudai, and Ohad Shamir

2: MLP-Mixer: An all-MLP Architecture for Vision. Ilya Tolstikhin, Neil Houlsby, Alexander Kolesnikov, Lucas Beyer, Xiaohua Zhai, Thomas Unterthiner, Jessica Yung, Andreas Steiner, Daniel Keysers, Jakob Uszkoreit, Mario Lucic, Alexey Dosovitskiy

3: Pay Attention to MLPs. Hanxiao Liu, Zihang Dai, David R. So, Quoc V. Le



**Summary Of The Paper:**

The authors construct a transformer that can memorize $N$ sequences of length $n$ and dimension $d$ that have permutation equivariance, with a transformer that has width 16 and depth $\tilde{\mathcal{O}}(n + \sqrt{N})$, with bit complexity $\tilde{\mathcal{O}}(\sqrt{N})$. Furthermore, their theory can show memorization for sequences without permutation equivariance using positional embeddings.

**Summary Of The Review:**

Overall, my scores are slightly on the negative side, mainly because (a) I would like to see how the attention module has been used in the proof sketch, and (b) I would like to see how the authors managed to improve upon Yun et al (2020a)'s construction. I would be very happy to discuss with the authors in the rebuttal period on the above questions.

---

> ### Author Response · Authors · 2022-11-18
> **Response to Reviewer Nzx7**
>
> We sincerely appreciate the time and effort that each reviewer spent on thoughtful reviews and comments. We address your comments below:
>
> > However, I feel that the use of the self-attention module in the construction hasn't been highlighted in the paper and is difficult to read from the proof in the appendix (please see more questions below).
>
> As per your comment, we significantly updated Section 3.3 in page 6 that contains the proof sketch. We mark all the revision in blue text and hope this revision and our following response would clarify how the attention module implements the contextual mapping in our proof.
>
> > One of the major questions that I have is whether we can use the result of [1] in two steps: first, we project each token in the $n$ length sequence into a single bit, and then further project the $n$ dimensional representation into a single bit.
>
> Thank you for the clarification question. We construct a sequence id as a linear combination of all values in the $n \times d$ input sequence. We exactly take the mentioned two-step approach compute the sequence id. However, the major challenge is in the second projection since the only allowed operation across tokens is attention. The result of [1] would not be directly applicable to the second projection because attention layers are permutation equivariant while feedforward layers used in [1] are not. The purpose of the contextual mapping step in our proof is to perform the second projection within the architectural constraint of Transformers.
>
> > A section similar to 3.1 in [1] would have helped a lot.
>
> We had our proof sketch in Section 3.3, which corresponds to Section 3.1 in [1]. We significantly update our draft to address your comment. We refer to the corresponding portion of the paper in our answers below.
>
> > (a) Can the authors explain the role of the self-attention module inside the construction? Can one simply use a feedforward network in place of the self-attention module? My reasoning is that after stage 1, you only need to store a single bit for each token in the sequence. Hence, is it possible to use a feedforward layer of dimension $n$ that "approximates maximum token id among the first coordinate values and outputs in the second coordinate"? \
> > If there is a difference, this can be used to compare attention models with MLP-only models introduced in [2, 3].
>
> The self-attention module presents only in stage 2 (contextual mapping) of our construction. Essentially, contextual mapping construct a single scalar (we call it sequence id) that identifies the sequence. Thus, a feedforward network in place of the self-attention module would work. However, there is a caveat.
>  * First, the number of parameter would be the same. Our construction uses $2n$ Transformer blocks of small constant width, resulting in $O(n)$ parameters. A single feedforward layer would also use $O(n)$ parameters to linearly project token ids into a sequence id.
>  * Second, a single feedforward layer would not be permutation equivariant. Although multiple feedforward layers would be able to simulate permutation equivariance by operating on values selected in decreasing order of magnitude, that would require much larger number of parameters ($O(n^2)$ parameters with $n$ feedforward layers each operating on a single coordinate). Thus, if the permutation equivariance is required, then the feedforward layers would be an inefficient alternative to the self-attention layers. However, when the equivariance across token is not desired, a single feedforward layer would work fine. And the same comment applies to [2, 3].
>
> For further details of attention module, we refer to the Section 3.3 in page 6 and 7, especially the newly added blue text.
>
> [1] On the Optimal Memorization Power of ReLU Neural Networks. Gal Vardi, Gilad Yehudai, and Ohad Shamir.
>
> [2] MLP-Mixer: An all-MLP Architecture for Vision. Ilya Tolstikhin, Neil Houlsby, Alexander Kolesnikov, Lucas Beyer, Xiaohua Zhai, Thomas Unterthiner, Jessica Yung, Andreas Steiner, Daniel Keysers, Jakob Uszkoreit, Mario Lucic, Alexey Dosovitskiy.
>
> [3] Pay Attention to MLPs. Hanxiao Liu, Zihang Dai, David R. So, Quoc V. Le.
>
> [4] Are transformers universal approximators of sequence-to-sequence functions? Chulhee Yun, Srinadh Bhojanapalli, Ankit Singh Rawat, Sashank Reddi, and Sanjiv Kumar.

---

> > ### Author Response · Authors · 2022-11-18
> > **Response to Reviewer Nzx7 (Continued)**
> >
> > > (b) Can the authors provide a discussion on how the contextual mapping used in Stage 2 improves over the selective shifting-based contextual mapping from Yun et al. (2020a)? Does the pre-processing in Stage 1 help reduce the overhead in Stage 2 for contextual mapping?
> >
> > We note our construction uses $O(n)$ layers/parameters. The selective-shifting-based contextual mapping from [4] requires $\left(\frac{1}{\delta}\right)^{dn}$ layers for shifting each grid cell of side length $\delta$. The pre-processing in stage 1 would reduce this to $\left(\frac{1}{\delta}\right)^{n}$ layers but it is still exponential.
> >
> > Actually, in the memorization setting, we may remove layers for grid cells without any data point. But the selective-shifting-based contextual mapping would still need $nN$ self-attention subblocks, which alone is already larger than the total number of required layers in our result. In contrast, our efficient contextual mapping uses $O(n)$ layers, improving all the above when $\delta, N > 1$. (The most practical settings fall into this regime.)
> >
> > The above answer is in the newly added text in Remark 3.6 in page 7 of the paper.
> >
> > > Also, can the authors provide some discussion on how their contextual mapping is capable of "incorporating sparse self-attention settings with minimal parameter overhead"?
> >
> > We show that the sparse attention with strictly weaker assumption than [5] is applicable to our contextual mapping with minimal overhead. We refer to Appendix C in page 26 in our updated draft.
> >
> > > (c) In figure 2, how do the authors observe a square root relation between training data size and model size? The plots seem very linear. In plot 2(b), if we remove the point (30000, 2.5e7) which could have been a noisy point, the plot will become more linear.
> >
> > We acknowledge that the trends in Figure 2 are more linear than square root dependence. We decided to withdraw our square root claim after inspection and update the paper accordingly. Instead, we only mention a subtle concavity in the low data regime.
> >
> > We note that the linear dependence is explainable through the bounded depth and bit complexity of the trained models. See Remark 3.3 in page 6 for more discussion. Our Theorem B.1 in page 24 and Theorem B.2 in page 25 have a formal statement of these cases.
> >
> > > (d) The experiments have been conducted with a fixed hyperparameter set (batch size, learning rate, etc.). That implies the plots in 2(a) and (b) could be highly dependent on the hyperparameters used. How do the plots look with a different set of hyperparameters?
> >
> > We agree that the trends may not be robust to hyperparameter choices. Unfortunately, we did not have enough time and resources to perform more experiments to check the robustness.
> >
> > From the smooth change of training errors in our heatmap and the granularity of our search space, we claim that the current linear trend at least provide complementary observation to our theory. We leave more detailed hyperparameter robustness test as a future work.
> >
> > > (e) Is there a relation between the entropy observed in the experiments in Figure 3 and the number of sequences $N$? (By theorem 3.1, there should be a square root dependence between bit complexity and $N$).
> >
> > There is a relation between our computed entropy and the model size as we compute entropy of a discrete probability distribution after taking softmax of a flattened parameter vector. Consequently, the number of sequences $N$ indirectly but strongly affect the entropy. After inspection, we decide to leave out Figure 3 and the corresponding discussion from the paper since it provides not much value in explaining bit complexity of trained networks.
> >
> > > (f) How do the plots in Figures 2 and 3 change with changes in the length of the sequences?
> >
> > We tested with a several different sequence length and observe the similar trend as in Figures 2 and 3. However, since the most of the data sequences are much shorter than padded sequences fed to the model, the model may have observed essentially the same length.
> >
> > > (g) The construction doesn't require the number of attention heads to change with a change in the number of sequences. Do the authors observe the same in their experiments, i.e. the behavior of the training loss doesn't depend on the number of attention heads used?
> >
> > We report a fixed number of attention heads in our experiment. When we experimented with a set of attention heads ({6, 12, 24}) with the same optimization setting, we did not notice a significant change in training loss.
> >
> > [4] Are transformers universal approximators of sequence-to-sequence functions? Chulhee Yun, Srinadh Bhojanapalli, Ankit Singh Rawat, Sashank Reddi, and Sanjiv Kumar.
> >
> > [5] O(n) connections are expressive enough: Universal approximability of sparse transformers. Chulhee Yun, Yin-Wen Chang, Srinadh Bhojanapalli, Ankit Singh Rawat, Sashank Reddi, and Sanjiv Kumar.

---

### Official Review · Reviewer_XAtG · 2022-10-25

**Confidence:** 3
**Correctness:** 3
**Technical Novelty And Significance:** 4
**Empirical Novelty And Significance:** Not applicable
**Recommendation:** 8

**Clarity, Quality, Novelty And Reproducibility:**

Clarity:
The paper is well-written and I enjoy reading it.

Quality:
The paper contains promising theoretical findings on Transformer models. I checked the detailed proof in the appendix twice and no major flaw is found.

Novelty:
Although the analysis is mainly based on the literature, the theoretical findings are new and interesting.

Reproducibility:
This is a theoretical paper and the proof contains enough details to help the readers to follow.


**Strength And Weaknesses:**

Strength:
This paper contains the technical in-depth results and sheds the light on explaining the efficiency of the transformer model.

Weakness:
1. The theoretical analysis is mainly an extension based on Vardi et al. (2022).
2. the current analysis requires $\mathcal{O}(n)$ transformer layers, which is a little bit too large for the common practice.

**Summary Of The Paper:**

This paper conducts the theoretical analysis of the memorization capacity of the Transformer model with finite machine precision and shows $\tilde{O}(d +n \sqrt{nN})$ model parameters are enough to memorize $N$ seq2seq samples with/without permutation equivariance assumption, where $d$ is the embedding dimension/model size and $n$ is the sequence length, which improves the results in previous literature (e.g., Yun et al., (2020ab) and Kratsios et al., (2022)). From the technical perspective, the authors consider a novel contextual mapping, which facilitates the analysis extending to the sparse attention settings. Moreover, the authors also use numerical experiments to verify the theoretical findings.

**Summary Of The Review:**

This paper proves that Transformers are capable of memorizing $N$ seq2seq samples with length up to $n$ with $\tilde{\mathcal{O}}(d + n + \sqrt{nN})$ parameters. The authors also use experiments to verify the theoretical findings.


Minor Issue:

Figure 2: From my point of view, figures 2(a)/(b) show the linear dependence even if in the beginning. The claim *..the square root relation between the training data size and the model size in the beginning...* is not well supported. I suggest the authors using separate figures to explicitly show square root dependence for the beginning stage.

Figure 3: The description of the x-axis in Figure 3 is missing and I suggest the authors adding some detailed explanation of this experiment, as based on the current presentation, it is confusing to me how the entropy of parameters is computed.

---

> ### Author Response · Authors · 2022-11-18
> **Response to Reviewer XAtG**
>
> We sincerely appreciate the time and effort that each reviewer spent on thoughtful reviews and comments. We address your comments below:
>
> > The theoretical analysis is mainly an extension based on Vardi et al. (2022).
>
> The main technical novelty that we claimed in our paper is the efficient contextual mapping construction. We explicitly reference [1] when we adopt their technique. As mentioned in Remark 3.6 on page 7 of our paper, adopting the known technique from [2] with some modifications would require $nN$ self-attention layers where $n$ is the sequence length and $N$ is the number of memorized sequences. This will dominate the parameter complexity obtainable from [2], hindering parameter-efficient memorization. In contrast, our construction uses $n$ self-attention layers. Please see the updated Remark 3.6 for the details. We update Remark 3.6 in our draft to clarify this point.
>
> We also claim the novelty in (1) formulating the memorization task for sequence data and (2) incorporating skip-connections in parameter-efficient memorization capacity.
>
> > The current analysis requires $O(n)$ transformer layers, which is a little bit too large for the common practice.
>
> We agree that $O(n)$ self-attention layers are more than what the Transformer models use in practice. As mentioned in the previous section, we improve even larger $O(nN)$ self-attention layers from [2]. We claim that removing the dependence on the number of data points $N$ is a significant improvement.
>
> We leave the question of tighter bounds on the number of self-attention layers required for memorization as future works. We conjecture that the number of self-attention layers for memorization may be further reduced, possibly (1) by making some data distribution assumptions, (2) by exploiting more attention heads or (3) by exploiting the representation capacity of softmax attention.
>
> > Figure 2: From my point of view, figures 2(a)/(b) show the linear dependence even if in the beginning. The claim ..the square root relation between the training data size and the model size in the beginning... is not well supported. I suggest the authors using separate figures to explicitly show square root dependence for the beginning stage.
>
> We acknowledge that the trends in Figure 2 are more linear than square root dependence. We decided to withdraw our square root claim after inspection and update the paper accordingly. Instead, we only mention a subtle concavity in the low data regime.
>
> We note that the linear dependence is explainable through the bounded depth and bit complexity of the trained models. See Remark 3.3 in page 6 for more discussion. Our Theorem B.1 in page 24 and Theorem B.2 in page 25 have a formal statement of these cases.
>
> > Figure 3: The description of the x-axis in Figure 3 is missing and I suggest the authors adding some detailed explanation of this experiment, as based on the current presentation, it is confusing to me how the entropy of parameters is computed.
>
> We compute entropy of a discrete probability distribution after taking softmax of a flattened parameter vector. After inspection, we decide to leave out Figure 3 and the corresponding discussion from the paper since it provides not much value in explaining bit complexity of trained networks.
>
> [1] On the Optimal Memorization Power of ReLU Neural Networks. Gal Vardi, Gilad Yehudai, and Ohad Shamir.
>
> [2] Are transformers universal approximators of sequence-to-sequence functions? Chulhee Yun, Srinadh Bhojanapalli, Ankit Singh Rawat, Sashank Reddi, and Sanjiv Kumar.

---

### Official Review · Reviewer_SL1m · 2022-10-28

**Confidence:** 2
**Correctness:** 4
**Technical Novelty And Significance:** 2
**Empirical Novelty And Significance:** 2
**Recommendation:** 8

**Clarity, Quality, Novelty And Reproducibility:**

The paper is well-written and clear. This paper is novel in the sense that it is the first to prove a memorization capacity for transformers, though the technical novelty may be limited. I find no errors in the proof so far, but I did not check every steps of the proof.

**Strength And Weaknesses:**

Strength: the memorization capacity of transformers is an interesting and important, though not surprising, topic to study. This paper is the first to prove the memorization capacity of transformers. The paper is well-written overall, with minor typos.

Weakness:  Two minor weakness are as follows. The technical novelty seems to be limited, as it mainly follows from Vardi et al. Also, there are some typos in the paper, e.g., in section 3.1 should $Y[:, k]$ and $Y[1, k]$ be using the same notation?

**Summary Of The Paper:**

This paper proves the memorization capacity of Transformer. The main technical theory is: given $N$ input-output pairs of sequence, this paper constructs a transformer that memorizes them with permutation equivariance, under slightly stronger assumptions. The mathematical tools used in this paper adopts the approach from Vardi et al., but with its own technical novelties. Finally, experiments with BERT validates the proposed theory.

**Summary Of The Review:**

The paper makes a solid contribution to first provide a memorization capacity result for transformers.

---

> ### Author Response · Authors · 2022-11-18
> **Response to Reviewer SL1m**
>
> We sincerely appreciate the time and effort that each reviewer spent on thoughtful reviews and comments. We address your comments below:
>
> > The technical novelty seems to be limited, as it mainly follows from Vardi et al.
>
> The main technical novelty that we claim in our paper is the efficient contextual mapping construction. We explicitly reference [1] when we adopt their technique. As mentioned in Remark 3.6 on page 7 of our paper, adopting the known technique from [2] with some modifications would require $nN$ self-attention layers where $n$ is the sequence length and $N$ is the number of memorized sequences. This will dominate the parameter complexity obtainable from [2], hindering parameter-efficient memorization. In contrast, our construction uses $n$ self-attention layers. Please see the updated Remark 3.6 for the details. We update Remark 3.6 in our draft to clarify this point.
>
> We also claim the novelty in (1) formulating the memorization task for sequence data and (2) incorporating skip-connections in parameter-efficient memorization capacity.
>
> > Also, there are some typos in the paper, e.g., in section 3.1 should $Y[:, k]$ and $Y[1, k]$ be using the same notation?
>
> Thank you for reporting the typo. Yes, we used them to mean the same entry of $Y$. We update our paper and use consistent notation to refer to entries in $Y$.
>
> [1] On the Optimal Memorization Power of ReLU Neural Networks. Gal Vardi, Gilad Yehudai, and Ohad Shamir.
>
> [2] Are transformers universal approximators of sequence-to-sequence functions? Chulhee Yun, Srinadh Bhojanapalli, Ankit Singh Rawat, Sashank Reddi, and Sanjiv Kumar.

---

### Author Response · Authors · 2022-11-18
**General Response to All Reviewers and Summary of Major Updates**

We would like to express gratitude to all the reviewers for giving us valuable and critical reviews. Your comments and questions shed a new perspective on the problem and allowed us to further improve our paper. We ask the reviewers to look into the updated version of our draft, for it reflects on the feedback of the reviewers. We emphasize our major update with blue color in the draft.

We are grateful that the reviewers enjoyed reading our paper (Reviewers XAtG and YgzV), found the theoretical findings and construction interesting (Reviewer XAtG and Nzx7), and acknowledged our contribution as solid (Review SL1m and YgzV). We also thank the reviewers for sharing their suggestions and concerns. Based on your comments, we improved our paper as follows:
* We updated Section 3.3 on page 6 to improve the description of the contextual mapping, the main technical contribution of our paper.
* We updated Remark 3.6 on page 7 to provide a better comparison against the selective-shifting-based contextual mapping in the prior work.
* We updated Section 5 on page 8 to improve the presentation of the experimental results. We correct our initial observation of square root trends. We describe the trend more precisely as a subtle concavity in the low data regime. Moreover, we leave out Figure 3 in the previous draft since it provides not much value in explaining the bit complexity of trained networks.
* We improved and provided more details on the proofs in Appendix A, B and C and the experimental details in Appendix F.
* We fixed inconsistent usage of indexing $Y$.

We thank reviewers again for the feedback. We hope our revision and response provides better presentation of our work and resolves concerns of all reviewers.

---

### Decision · Program_Chairs · 2023-01-20

**Decision:**

Accept: poster

**Justification For Why Not Higher Score:**

Limited technical novelty.

**Justification For Why Not Lower Score:**

New result on memorization for Transformer architecture which is of interest to the community.

**Metareview: Summary, Strengths And Weaknesses:**

All the reviewers find the results interesting and useful towards improving our understanding of transformers. Main concerns were around presentation of proof sketch and comparison to existing works. Authors addressed both of these concerns in response and I am happy to suggest acceptance. I encourage authors to include all the reviewer suggestions while preparing final version.

Pros -
1. New memorization result for Transformers.
2. Analysis techniques to handle weight sharing.

Cons-
1. Limited technical novelty with the work borrowing lot of techniques from existing results.
2. Lack of clear presentation of proof sketch and comparisons.

**Note From Pc:**

if the above contains the word "oral" or "spotlight" please see: "oral" presentation means -> notable-top-5% and "spotlight" means -> notable-top-25%. As stated in our emails, we are disassociating presentation type from AC recommendations

**Summary Of Ac-Reviewer Meeting:**

The main concerns of most reviewers were how the result and techniques compare to existing works. Authors addressed this well in rebuttal and most reviewers are happy with the response. One reviewer gave a score of 5, however they didn't attend the discussion stage. I think all their concerns were well addressed in the response.